**Processing and performance of topobathymetric LiDAR data for geomorphometric and morphological classification in a high-energy tidal environment in the coastal zone**

**M.S. Andersen[1], Á. Gergely[1], Z. Al-Hamdani[2], F. Steinbacher[3], L.R. Larsen[4], V.B. Ernstsen[1]**

[1] Department of Geosciences and Natural Resource Management, University of Copenhagen, Denmark

[2] Geological Survey of Denmark and Greenland, Denmark

[3] Airborne Hydro Mapping GmbH, Austria

[4] NIRAS, Denmark

Correspondence to: V. B. Ernstsen (vbe@ign.ku.dk)

**Abstract**

The transition zone between land and water is difficult to map with conventional geophysical systems due to shallow water depth and often harsh environmental conditions. The emerging technology of airborne topobathymetric Light Detection And Ranging (LiDAR) is capable of providing both topographic and bathymetric elevation information, using only a single green laser, resulting in a seamless coverage of the land-water transition zone. However, there is no transparent and reproducible method for processing green topobathymetric LiDAR data into a Digital Elevation Model (DEM). The general processing steps involve data filtering, water surface detection and refraction correction. Specifically, the procedure of water surface detection, solely using green laser LiDAR data, has not previously been described in detail. The aim of this study was to fill this gap of knowledge by developing a step-by-step procedure for modelling the water surface using the green laser LiDAR data. The detailed description of the processing method augments its reliability, makes it user friendly and repeatable.

A DEM was obtained from the processed topobathymetric LiDAR data collected in
spring 2014 from the Knudedyb tidal inlet system in the Danish Wadden Sea. The
vertical accuracy of the LiDAR data is determined to ±8 cm at a 95% confidence level,
and the horizontal accuracy is determined as the mean error to ±10 cm. The LiDAR
technique is found capable of detecting features with a size of less than 1 m$^2$. The
derived high resolution DEM was applied for detection and classification of
geomorphometric and morphological features in the study area. Initially, stage (or
elevation in relation to tidal range) was used to divide the area of investigation into the
different tidal zones, i.e. subtidal, intertidal and supratidal. Subsequently, a combination
of statistical neighbourhood analyses (Bathymetric Positioning Index, moving average
and standard deviation) with varying window sizes, combined with the first derivative
slope and the area/perimeter-ratio were used to identify and characterise morphometric
units. Finally, these morphometric units were classified into six different types of
morphological features (i.e. subtidal channel, intertidal flat, intertidal creek, linear bar,
swash bar and beach dune). The developed classification method is adapted and applied
to a specific case, but it can be transferred to other cases and environments.

## 18      1    Introduction

The coastal zone is under pressure from human exploitation in many and various ways.
Many large cities are located near the coast, and they grow gradually with the increase
in worldwide population and urbanization. Many industrial activities take place in close
vicinity to the coast, e.g. fishery, construction, maintenance dredging for safety of
navigation, and mining for raw materials. The coastal zone also provides the setting for
many recreational and touristic activities, such as sailing, swimming, hiking, diving and
surfing. In addition to human exploitation, climate change also poses a future threat
with a predicted rising sea level and increasing storm intensity and frequency, expected
to cause erosion and flooding in the coastal zone (Mousavi et al., 2011). All these
pressures and different interests underpin the societal need for high resolution mapping,
monitoring, and sustainably managing of the coastal zone.
The transition zones between land and water have been difficult or even impossible to
map and investigate in high spatial resolution due to the challenging environmental

conditions. The airborne near-infrared (NIR) Light Detection and Ranging (LiDAR) is a technique often used for measuring high-resolution topography, however, NIR laser is incapable of measuring bathymetry due to the absorption and reflection of the laser light at the water surface. Traditionally, high-resolution bathymetry is measured with a multibeam echosounder (MBES) system mounted on a vessel, but it does not cover the bathymetry in the shallow water due to the vessel draft limitation.

NIR LiDAR and MBES are applied in different environments; however, the data are very similar and the processed high-resolution topography/bathymetry are both often captured in a Digital Elevation Model (DEM). The processed DEM may be applied for various purposes, e.g. for geomorphological mapping. Previous studies classifying morphology in either terrestrial or marine environments have been performed numerous times (Al-Hamdani et al., 2008; Cavalli and Marchi, 2008; Höfle and Rutzinger, 2011; Ismail et al., 2015; Kaskela et al., 2012; Lecours et al., 2016; Sacchetti et al., 2011). These classification studies generally focus on either the marine or the terrestrial environment, and they do not cover the small-scale morphology in the shallow water at the land-water transition zones, due to the challenges of collecting data in these high-energy environments. A new generation of airborne green topobathymetric LiDAR enables high resolution measurements of both topography and shallow bathymetry, and for that reason it is specifically suited to map the land-water transition zone (Guenther, 1985; Jensen, 2009; Pe'eri and Long, 2011). The potential of merging morphological classifications of marine and terrestrial environments enables a holistic approach for managing the coastal zone.

Topobathymetric LiDAR is based on continuous measurements of the distance between an airplane and the ground/seabed. The distance (or range) is calculated by half the travel time of a laser beam, going from the airplane to the surface of the earth and back to the airplane. The wavelength of the laser beam is in the green spectrum, usually 532 nm, since this wavelength is found to attenuate the least in the water column, resulting in the largest penetration depth of the laser (Jensen, 2009). In literature, topobathymetric LiDAR data is sometimes referred to as either bathymetric LiDAR or Airborne LiDAR bathymetry (ALB). These are just different terms with the same meaning, and in this paper, topobathymetric LiDAR is preferred, since it describes the system's ability to simultaneously measure bathymetry as well as topography.

A single laser beam may encounter many targets of varying nature on its way from the
airplane and back again, and different processes are influencing the laser beam
propagation through air and water. First, the laser beam may be reflected by targets in
the air, such as birds or dust particles, and these can show up as LiDAR reflection points
in the space between the airplane and the surface. When encountering water, the speed
of the laser decreases from $3 \times 10^8$ ms$^{-1}$ to e.g. $2.25 \times 10^8$ ms$^{-1}$ in 10°C freshwater or
e.g. $2.24 \times 10^8$ ms$^{-1}$ in 10°C saltwater of 30 PSU (Millard and Seaver, 1990).
The changing speed of the laser beam also affects the direction of the laser beam when
penetrating the water surface with an angle different from nadir (Fig. 1) (Guenther,
2007; Jensen, 2009). The laser beam will be refracted according to Snell's Law
(Mandlburger et al., 2013):
$$\frac{\sin \alpha_{air}}{\sin \alpha_{water}} = \frac{c_{air}}{c_{water}} = \frac{n_{water}}{n_{air}} \tag{1}$$
where $\alpha_{air}$ is the incidence angle of the laser beam relative to the normal vector of the
water surface and $\alpha_{water}$ is the refraction angle in water. $n_{water}$ and $n_{air}$ are the
refractive indices of water and air, respectively (Mandlburger et al., 2013).
The penetration depth in water is limited by the attenuation of the laser beam. Water
molecules, suspended sediment and dissolved material all act on the laser beam by
absorption and scattering, resulting in substantial reduction in power as the signal
propagates into the water (Guenther, 2007; Mandlburger et al., 2013; Steinbacher et al.,
2012). The laser beam also diverges in the water column, resulting in a wider laser
beam footprint (Guenther et al., 2000), and this effect reduces the resolving capability of
fine-scale morphology the deeper the laser beam penetrates.
The returned signal is represented as a distribution of energy over time, also called the
'full-waveform' (Alexander, 2010; Chauve et al., 2007; Mallet and Bretar, 2009). The
peaks in the full-waveform are detected as individual targets encountered by the
propagating laser beam. If the laser hits two targets with a small vertical difference,
such as a water surface and seabed in very shallow water, then the two peaks in the full-
waveform may merge together, resulting in the detection of only one target (Fig. 1).
This results in a detection minimum of successive returns from a single laser pulse, and
the vertical distance within this minimum is referred to as the 'dead zone' (Mandlburger

et al., 2011; Nayegandhi et al., 2009). The dead zone is a clear limitation to the LiDAR measurements, which is an important parameter to consider in very shallow water, such as intertidal environments.

The raw LiDAR measurements are spatially visualized as a point cloud, with each point representing an individual target. The point cloud must be piped through a series of steps before it can take shape as a DEM. Most of the processing steps required to process raw topobathymetric LiDAR data to a DEM are similar to the processing steps of topographic LiDAR data (Huising and Gomes Pereira, 1998). However, additional processing steps are required for topobathymetric LiDAR data due to the refraction of the laser beam at the water surface. All submerged LiDAR points have to be corrected for the refraction, but in order to do so, the water depth must be known for each point. This sets a requirement of modelling the water surface before the refraction correction can be performed. The general processing procedure is well defined; however, there is no standard or universal approach for how to deal with these steps. LiDAR companies have their workflows, but the specific steps in their workflow are usually hidden, which make them non-repeatable.

In particular, there is no definitive method for detecting a water surface from green topobathymetric LiDAR data. Often the water surface is detected from simultaneous collection of green and NIR LiDAR measurements, where the green laser reflects from the seabed and the NIR laser reflects from the air-water interface, and the NIR laser data are then used to detect and model the water surface (Allouis et al., 2010; Collin et al., 2008; Guenther, 2007; Parker and Sinclair, 2012). The use of NIR LiDAR data for water surface detection has been applied in several studies. For instance, Hofle et al. (2009) proposed a method for mapping water surfaces based on the geometrical and intensity information from NIR LiDAR data. Su and Gibeaut (2009) classified water points from NIR LiDAR based on point density, intensity and altitude. They identified the shoreline based on the large sudden decrease in NIR LiDAR intensity values when going from land to water. Brzank et al. (2008) used the same three variables (point density, intensity and altitude) in a supervised fuzzy classification to detect the water surface in a section of the Wadden Sea. Another study in the Wadden Sea by Schmidt et al. (2012) used a range of geometric characteristics as well as intensity values to classify water points from NIR LiDAR data.

The capability of NIR LiDAR data for water surface detection is thus well documented. However, deriving all the information (seabed and water surface) from a single green LiDAR dataset would be a more effective solution for water surface detection, with respect to the financial expenses and for the difficulties of storing and handling often very large amounts of data. For this purpose, the Austrian LiDAR company RIEGL have developed a software, *RiHYDRO* (RIEGL, 2015), in which it is possible to model the water surface in a two-step approach: 1) Classification of water surface points based on areas with two layers (water surface and seabed) and extending the classification to the entire water body, and 2) Generation of a geometric gridded water surface model for each flight swath based on the classified water surface points. However, RiHYDRO is commercial software, and thus the algorithms, which form the basis of the classification and water surface modelling, are not publicly available. Other software packages, such as *HydroFusion* (Optech, 2013) and *LiDAR Survey Studio* (Leica, 2015), also proclaim to have incorporated methods for the entire data processing workflow, but the algorithms in these software packages are also closed and cannot be accessed by users.

Only few research studies have investigated the potential of water surface detection from green LiDAR data. Guenther et al. (2000) even regarded water surface detection from green LiDAR data as unacceptable and they justified it with two fundamental issues: 1) No water surface returns are detected in the dead zone, and 2) Uncertainty of the water surface altitude, because the green water surface returns are actually a mix of returns from the air/water interface and from volume backscatter returns, and they are generally found as a cloud of points below the water surface. Mandlburger et al. (2013) addressed the second issue by comparing the water surface points of NIR and green LiDAR data, and they concluded that it is possible to derive the water surface altitude from the green LiDAR data with sub-decimetre vertical precision relative to a reference water surface derived by the NIR LiDAR data. However, their work addressed only the determination of the water surface altitude, without going into detail on the actual procedure of modelling the water surface. An approach for modelling the water surface from green LiDAR data was presented by Mandlburger et al. (2015), who did their study in a riverine environment with only few return signals from the water surface. Their method was based on manual estimates of the water level in a series of river cross sections, after which interpolation between the cross sections filled out the gaps with no

water surface points to derive a continuous water surface model. The vertical accuracy of the detected water surface was evaluated by statistical comparison against water surface points from a terrestrial laser scanner, resulting in a root mean square error of ±3.3 cm.

Published literature that deals with water surface modelling/detection procedure in the coastal zone based solely on green laser Lidar data are very few and the procedure for LiDAR data processing to reach this goal is not clearly explained.

The aim of this study was to investigate the potential of improving the processing procedure of green LiDAR data for generating DEMs in tidal coastal environments characterised by land-water transition zones, and of improving the classification of morphological units in such environments. More specifically, the objectives were:

1. To develop a robust, repeatable and user friendly processing procedure of raw green LiDAR data for generating high resolution DEMs in land-water transition zones.
2. To quantify the accuracy and precision of the green LiDAR data based on object detection.
3. To automatically classify morphological units based on morphometric analyses of the generated DEM.

The investigations were based on studies undertaken in a section of the Knudedyb tidal inlet system in the Danish Wadden Sea.

## 2   Study area

The Knudedyb tidal inlet system is located between the barrier islands of Fanø and Mandø in the Danish Wadden Sea (Fig. 2A). The tidal inlet system is a natural environment without larger influence from human activity. The tides in the area are semi-diurnal, with a mean tidal range of 1.6 m, and the tidal prism is in the order of $175 \cdot 10^6$ m$^3$ (Pedersen and Bartholdy, 2006). The main channel is approximately 1 km wide and with an average water depth of approx. 15 m (Lefebvre et al., 2013).

The study site is an elongated 3.2 km$^2$ (0.85 × 4 km) section of the Knudedyb tidal inlet system (Fig. 2B). The section is located perpendicular to the main channel and stretches across both topography and bathymetry. The study site extends towards north into an

area on Fanø with dispersed cottages (Fig. 2C). The most prominent morphological features within the study site include beach dunes (Fig. 2D), small mounds (Fig. E), swash bars (Fig. 2F-G) and linear bars (Fig. 2H).The quality of the LiDAR data were validated at two sites along Ribe Vesterå River (Fig. 2I-J):

- Validation site 1 is a cement block with a size of 2.50×1.25×0.80 m located on land next to the mouth of Ribe Vesterå River (Fig. 2I). The block was used for assessing the accuracy and precision of the LiDAR data.

- Validation site 2 is a steel frame with a size of 0.92×0.92×0.30 m located in the river with the surface just below the water surface (Fig. 2J). The frame was used for precision assessment, and for testing the feature detection capability of the LiDAR system. According to the hydrographic survey standards presented by the International Hydrographic Organization (IHO, 2008), cubic features of at least 1 m$^2$ should be detectable in Special Order areas, which are areas with very shallow water as in the study site.

## 3   Methods

### 3.1   Surveys and instruments

LiDAR data and ortophotos were collected by Airborne Hydro Mapping GmbH (AHM) during two surveys on 19 April 2014 and 30 May 2014.

On 19 April 2014, validation sites 1 and 2 were covered for accuracy and precision assessment of the LiDAR data by object detection of the block and the frame (for location see Fig. 2). The block was covered by 7 swaths retaining 227 LiDAR points from the block surface. The frame was covered by 4 swaths retaining 46 LiDAR points from the surface of the frame. Ground control points (GCPs) were measured for the four corners of the block with accuracy better than 2 cm using a Trimble R8 RTK GPS. Measurements were repeated three times and averaged to minimize errors caused by measurement uncertainties. GCPs were also collected for the frame; however, during the LiDAR survey the frame experienced an unforeseen intervention by local fishermen using the frame as fishing platform. Therefore, the frame is only used to assess the

deviation between the LiDAR points (the precision), and not to assess the deviation between the LiDAR points and GCP's (the accuracy).

On 30 May 2014, the study site was covered by 11 swaths, which were used for generating the DEM. Low tide was -1 m DVR90, measured at Grådyb Barre, approx. 20 km NW of the study site.

The weather conditions were similar during the two surveys, with sunny periods, average wind velocities of 7-8 m/s (DMI, 2014) and approx. 0.5 m wave heights coming from NW, measured west of Fanø (DCA, 2014). The wave heights in the less exposed Knudedyb tidal inlet was observed in the LiDAR data to 0.2-0.3 m. Overall, both days constituted good conditions for topobathymetric LiDAR surveys.

In both surveys, LiDAR data were collected with a RIEGL VQ-820-G topobathymetric airborne laser scanner. The scanner is characterized by emitting green laser pulses with 532 nm wavelength and 1 ns pulse width. It has a very high laser pulse repetition rate of up to 520,000 Hz, and a beam divergence of 1 mrad creates a narrow laser beam footprint of 40 cm diameter at a flying altitude of 400 m (RIEGL, 2014), which was the actual flying altitude during the surveys. The high repetition rate and narrow footprint makes it well suited to capture fine-scale landforms (Doneus et al., 2013; Mandlburger et al., 2011; RIEGL, 2014). An arc shaped scan pattern results in a swath width of approx. 400 m (at 400 m flying altitude), while maintaining an almost constant 20° (±1°) incidence angle of the laser beam when it penetrates the water surface (Niemeyer and Soergel, 2013). The typical water depth penetration of the laser scanner is 1 Secchi disc depth.

For each returned signal, the collected LiDAR data contained information of x, y and z, as well as a GPS time stamp and values of the amplitude, reflectance, return number, attribute and laser beam deviation (RIEGL, 2012).

## 3.2 Processing raw topobathymetric LiDAR data into a gridded DEM

The essential processing steps, which are standard procedure when processing topobathymetric LiDAR data, were followed to produce a DEM in the study area. These steps included:

1. Determination of flight trajectory.

2. Boresight calibration: Calculating internal scanner calibration.

3. Collecting topobathymetric LiDAR data.

4. Swath alignment based on boresight calibration: The bias between individual swaths was minimized.

5. Filtering: The raw data contained noise located both above and below ground, which needed to be filtered from the point cloud.

6. Water surface detection: A water surface had to be established in order to correct for refraction in the following step.

7. Refraction correction: All the points below the water surface were corrected for the refraction of the laser beam.

8. Point cloud to DEM: The points were transformed into a surface representing the real world topography and bathymetry.

Step 1 and 2 were performed prior to the LiDAR survey. The different instruments (LiDAR, IMU and GPS) were integrated spatially by measuring their position relative to each other, when mounted on the airplane, and temporally by calibrating their time stamps.

Step 3 was the actual LiDAR survey and step 4 was the initial processing step after the LiDAR survey. The bias between the swaths was minimized in the software RiPROCESS (RIEGL LMS) by automatically searching for planes in each swath and then matching the planes between the swaths.

Step 5-8 represents the processing of the point cloud into a DEM. The methods involved in these steps are the main focus in this work and they are described in detail in the following sub-sections. Each swath was pulled individually through the processing workflow to account for the continually changing water level in the study area due to tides.

## 3.2.1 Filtering

The raw LiDAR data contained noise in the air column originating from the laser being scattered by birds, clouds, dust and other particles, and noise was also appearing below the ground/seabed (Fig. 3A-B). This noise had to be filtered before further processing. The filtering process involved both automatic and manual filtering.

1. Automatic filtering
The automatic filtering was carried out in HydroVish (AHM) with the tool *Remove flaw*
*echoes* (Fig. 3C). The filtering tool was controlled by three variable parameters: search
radius, distance and density. The search radius parameter specified the radius of a
sphere in which the distance and density filters were utilized. The distance parameter
rejected a point, if it was too far from any other point within the sphere. The density
parameter specified the lower limit of points within the sphere. The automatic filter
iterated through all the points in the point cloud.
In order to identify the best settings of the three parameters, a sensitivity analysis was
performed on three data fragments representing different natural environments in the
Knudedyb tidal inlet system: a fragment in the flood channel, one on the tidal flat and a
fragment with vegetation. The outcome of the filtering was compared for different
settings to decide the most suitable settings to use for filtering the whole study area. It
was not possible to reach a specific setting, which would be optimal for all the different
environments. Particularly, the deeper bathymetric parts contained more widely
dispersed points, which were easily rejected by the filter. The analyses with different
settings also showed that two layers of noise close to the ground, both above and below,
were very difficult, if not impossible, to reject with this automatic filtering method. The
settings were selected so that a minimum of valid points were rejected by the automatic
filter. The settings were: Search radius = 1 m, distance = 0.75 m and density = 4.
2. Manual filtering
The remaining noise was manually filtered in the software Fledermaus (QPS) (Fig. 3D).
The filtered point cloud (with water points) was used in the following step to detect the
water surface. Meanwhile, a copy of the data were undergoing additional manual
filtering, removing all the water points (Fig. 3E). After this final filtering step, there
were only points representing topography, bathymetry, vegetation and man-made
structures left in the dataset.
**3.2.2  Water surface detection**
The water surface detection was based on determining the water surface *altitude* and the
water surface *extent*. The water surface altitude was determined based on the water

surface points and the extent was determined by extrapolating the water surface until it intersected the surface of the topography. Two assumptions about the water surface were made:

1. The water surface was horizontal. This was a simplification of the real world. Tidal processes and wind- and wave-setup may cause the water surface to be sloping, and the water is often topped by more or less significant wave action. A linear fit through the water surface LiDAR points along the main channel, showed a changing water level of 0.13 m over a distance of 400 m, corresponding to a $0.325 \times 10^{-3}$ (0.019 deg.) sloping water surface. A similar fit through the LiDAR points along the flood channel showed a slope of $0.156 \times 10^{-3}$ (0.009 deg.). The maximum wave heights observed in the main channel were 20-30 cm. Based on the moderate slope of the water surface and relatively low wave height, the water surface was assumed flat. This assumption is deemed error prone, but at the time of this study, it was currently our best estimate.

2. The study area contained water bodies with two different water levels: One represented the water level in the main channel and the other represented the water level in the flood channel. This was also a simplification, as the tidal flat contained small ponds with potentially different water levels. However, almost all of these ponds contained no LiDAR points of the water surface, which means that the water depth in the ponds must have been within the limitation of the dead zone. Therefore, it was impossible to detect individual water surfaces in the ponds.

A series of processing steps were performed to detect the water surface. The first step was to extract a *shallow surface* and a *deep surface* from the filtered point cloud (with water points) in Fledermaus (Fig. 3F). Both surfaces consisted of $0.5 \times 0.5$ m cells, and the altitude of each cell was equal to the highest point within the cell (shallow surface) and the lowest point within the cell (deep surface), respectively. The shallow surface should then display the topography along with the water surface, whereas the deep surface should display the topography and the seabed (as long as the seabed was detected by the laser). It is worth noting, that the extraction of the shallow surface and the deep surface have nothing to do with the final DEM, as they are just intermediate steps performed for the water surface detection.

The following steps were focused on the shallow surface to determine the altitude of the water surface (Fig. 3G). First, the shallow surface was down-sampled to a surface with a cell size of $2 \times 2$ m, and the new cells were populated with the maximum altitude of the input cells. The down-sampling was done for smoothing the water surface, and thereby eliminating most of the outliers. The exact cell size of $2 \times 2$ m, as well as populating them with the maximum value, was chosen based on the work by Mandlburger et al. (2013). They compared water surface detection capability between green LiDAR data, collected with the same RIEGL-VQ-820-G laser scanner, and NIR LiDAR data, which was assumed to capture the true water surface. They found that the green LiDAR generally underestimated the water surface level, but that reliable results were achieved by increasing the cell size and only taking the top 95-100% of water points into account. According to their work, it was assumed that placing the water surface on the highest points in 2 m cells provided a good estimate of the true water level. However, based on their results it could be expected that the water surface level in this case would be underestimated in the order of 2-4 cm.

The water covered areas in the main channel and the flood channel were manually extracted from the newly down-sampled raster surface. The average altitude of the cells was calculated individually in each area, and these values constituted the water surface levels in the main channel and in the flood channel, respectively.

Hereafter, the extent of the water surfaces was determined (Fig. 3H). Two horizontal water surfaces were created in the flood channel and the main channel with a cell size of $0.5 \times 0.5$ m and cell values equal to the determined water surface altitudes in each region. The high spatial resolution of 0.5 m cells was chosen to produce a detailed water surface along the edges of the land-water transition. It also made the calculations in the following step straightforward, because the resolution was similar to that of the deep surface. The deep surface cell altitudes were subtracted from the water surface altitude and all cells with resulting negative values were discarded from the water surface. Thereby, all the water surface cells which were below the deep surface were discarded. All the cells above the deep surface were expected to represent the two water surfaces. Thereby, two water surfaces were created; one in the main channel and one in the flood channel.

### 3.2.3 Refraction correction

The refraction correction of all the points below the water surfaces was calculated in HydroVish (AHM). The input parameters were the filtered point cloud (without water points), the derived water surfaces and the trajectory data of the airplane. These were all converted to F5 file format to allow import into HydroVish (AHM). The refraction correction was calculated automatically for each point based on the water depth, the incident angle of the laser beam, and the refracted angle according to Snell's Law (Eq. 1 and Fig. 3I).

### 3.2.4 Point cloud to DEM

After iterating through the processes of filtering, water surface detection and refraction correction for all the individual swaths, the LiDAR points of all swaths were combined. The transformation from point cloud into a DEM was performed with ArcGIS (ESRI) software. The DEM was created as a raster surface with a cell size of $0.5 \times 0.5$ m, and each cell was attributed the average altitude of the points within the cell-boundaries. It was chosen to make the resolution of the DEM lower than the laser beam footprint size (i.e. 40 cm), due to the inaccuracies arising from attributing smaller cells with measured altitude values spanning across a larger area. Furthermore, the 0.5 m cell size was chosen to get as high resolution as possible without making any significant interpolation between the measurements. In this way, each cell represented actually measured altitudes instead of interpolated values. However, there were still very few gaps of individual cells with no data in the resulting raster in areas with relatively low point density. Despite of the general intention of avoiding interpolation it was chosen to populate these cells with interpolated values to end up with a full DEM coverage (except for the bathymetric parts beyond the maximum laser penetration depth). The arguments for interpolation were that 1) the interpolated cells were scattered and represented only 1.7 % of all the cells, 2) they were found primarily on the tidal flat where the slope is generally less than 1°, meaning that the altitude difference from one cell to a neighbouring cell is usually less than 1 cm, and 3) the general point density in most of the study area was so high that the loss of information by lowering the DEM resolution would represent a larger sacrifice than interpolating a few scattered cells. The interpolation was performed by assigning the average value of all neighbouring cells to

the empty cells. The final DEM was thereby fully covering the topography, and the
bathymetry was covered down to a depth equal to the maximum laser penetration depth.
**3.3   Accuracy and precision of the topobathymetric LiDAR data**
The term a*ccuracy* refers to the difference between a point coordinate (in this case a
LiDAR point) compared to its "true" coordinate measured with higher accuracy, e.g. by
a total station or a differential GPS; while the term *precision* refers to the difference
between successive point coordinates compared to their mean value, i.e. the
repeatability of the measurements (Graham, 2012; Jensen, 2009; RIEGL, 2014).
Two "best-fit planes" based on the LiDAR points on the block and the frame surfaces
were established with the *Curve Fitting tool* in MATLAB (MathWorks). We propose
the use of these two planes to give an indication of the relative precision of the LiDAR
measurements.
Another best-fit plane was established based on the block GPS measurements, and this
plane was regarded as the "true" block surface for assessment of the accuracy of the
LiDAR measurements. The established planes were described by the polynomial
equation:
$z(x, y) = a + bx + cy$                                                   (2)
where $x$, $y$ and $z$ are coordinates and a, b and c are constants. Inserting x and y
coordinates for the LiDAR surface points in Eq. (3) led to a result of the corresponding
altitude (z) as projected on the fitted plane. The difference between the altitude of the
LiDAR point and the corresponding altitude on the fitted plane was used as a measure
of the vertical accuracy (for the GCP fitted plane of the block) and the vertical precision
(for the LiDAR point fitted plane of the block and the frame). Statistical measures of the
standard deviation ($\sigma$), mean absolute error ($E_{MA}$), and root mean square error ($E_{RMS}$)
were calculated by:
$$\sigma = \sqrt{\frac{\sum(z_i - z_{plane})^2}{n-1}}$$                                       (3)
$$E_{MA} = \frac{\sum|z_i - z_{plane}|}{n}$$                                           (4)

$$E_{\text{RMS}} = \sqrt{\frac{\sum(z_{\text{i}} - z_{\text{plane}})^2}{n}} \hspace{4cm} (5)$$

where $z_i$ is the altitude of the measured LiDAR points, $z_{\text{plane}}$ is the corresponding altitude on the best-fit plane, and $n$ is the number of LiDAR points. The vertical accuracy and precision were determined at a 95% confidence level based on the accuracy standard presented in *Geospatial Position Accuracy Standards Part 3: National Standard for Spatial Data Accuracy* (NSSDA) (FGDC, 1998):

$$CI_{95\%} = E_{\text{RMS}} \cdot 1.96 \hspace{4cm} (6)$$

The horizontal accuracy was determined as the horizontal mean absolute error ($E_{\text{MA,xy}}$) based on the horizontal distances between the block corners, measured with RTK GPS, and the best approximation of the block corners derived from the LiDAR points of the block surface. The minimum distance between a block corner and the perimeter of the LiDAR points was regarded as the best approximation. Hereafter $E_{\text{MA,xy}}$ was calculated as the average of the four corners.

## 3.4 Geomorphometric and morphological classifications

The processed DEM was applied in two classification analyses; first a *geomorphometric* classification and then a *morphological* classification. Both were based on the DEM and derivatives of the DEM, but they differentiated by the resulting classification classes, which showed 1) Surface geometry and 2) Surface morphology.

Geomorphometric classification analysis

The tool Benthic Terrain Modeler (BTM) (Wright et al., 2005) was used for the geomorphometric classification. The tool is an extension to ArcGIS Spatial Analyst, originally used for analysing MBES data (Diesing et al., 2009; Lundblad et al., 2006; Rinehart et al., 2004). The BTM classification tool uses fine- and broad scale Bathymetric Positioning Indexes (BPIs) (Verfaillie et al., 2007) in a multiple scale terrain analysis to classify fine- and broad scale geometrical features. The BPIs are measures of the altitude of a cell compared to the altitude of the surrounding cells within the determined scale (radius) size. Positive BPI values indicate a higher altitude than the neighbouring cells and negative BPI values indicate a lower altitude than the neighbouring cells. For instance, a BPI value of 100 corresponds to 1 standard deviation

and a value of -100 corresponds to -1 standard deviation. BPI values close to zero are
derived from flat areas or from constant slopes.
The altitude of the DEM was exaggerated 10 times before the classification, to enable
the BTM to detect the shapes of the landscape. The fine- and broad scales were
determined based on the BPI results for different radius sizes. The best results were
obtained from a broad scale BPI of 100 m radius and a fine scale BPI of 10 m radius.
The fine- and broad scale BPIs were used, together with DEM derived slopes to classify
the investigated area into the geomorphometric classes: Small-scale crests, large-scale
crests, depressions, slopes and flats (Fig. 4). The classification classes were decided
based on previous studies using the BTM classification tool with success (Diesing et al.,
2009; Lundblad et al., 2006). The thresholds for the fine- and broad scale BPIs were in
previous studies often defined as 1 standard deviation (Lundblad et al., 2006; Verfaillie
et al., 2007), however, thresholds of 0.5 standard deviations have also previously been
applied (Kaskela et al., 2012). We used a low threshold of 0.5 standard deviations due
to the generally very gentle variations in the landscape geometry of the tidal inlet
system. We defined the threshold between slopes and flats as 2°. This definition was a
compromise between detecting as many slopes as possible but avoiding too many "false
slopes" being detected along the swath edges, which seemed to be a consequence of
lower precision at the outer beams of the swath, as well as differences between
overlapping swaths.

21          Morphological classification analysis

A morphological classification was developed for the purpose of delineating classes of
actual morphological features in the study area. This classification was built partly on
different neighbourhood analyses and slopes derived from the DEM, and partly on the
local tidal range. Large scale crests from the geomorphometric classification were also
incorporated in the analysis.Figure 5 describes the steps performed in ArcGIS, which
led to the classification of 6 morphological classes: Swash bars, linear bars, beach
dunes, intertidal flats, intertidal creeks and subtidal channels. All the criteria for
defining a particular morphological class had to be fulfilled for a cell to be classified
into that class.
33 years of continuous measurements of the water level at Havneby on Rømø, 25 km
south of the study area, shows a mean low water level of -0.94 m (DVR90) and a mean
high water of 0.94 m (DVR90) (Klagenberg et al., 2008). Although the tidal range in
Knudedyb is probably slightly different, it is the best estimate for the study site.
Therefore, these water levels were used to separate between the supratidal, intertidal and
subtidal zones.
Subtidal channels were defined as everything below the mean low water, which is -0.94
m. A "smooth DEM" was created, in which each cell of the original DEM was assigned
the average altitude value of its surrounding cells in a window size of 100x100 m. The
result was subtracted from the original DEM, creating an Elevation Change Model
(ECM), which made it possible to extract information about the deviation of the cells in
the DEM compared to its surrounding cells. The principle is similar to the BPI, and
again the purpose was to locate cells, with a higher/lower altitude than its surrounding
cells. Positive values were higher cells and negative values were lower cells. Certain
thresholds were found suitable for classifying beach dunes (> 0.8 m) and intertidal
creeks (< -0.3 m). These two classes were furthermore classified into their respective
tidal zones (supratidal and intertidal) based on the altitude. Intertidal flats were
classified by low slope values (< 1°) of a down-sampled 2 m DEM (each down-sampled
cell was assigned the mean value of its 4x4 original cells). Moreover, to be classified as
a flat, the ECM has to be within ±10 cm to avoid any incorrect intertidal flat
classification of flat crests on top of bars or flat bottoms inside creeks or channels. The
BTM classification class "large-scale crests" is used as an input, since it is found to
capture bar features. However, the thresholds used in the BTM classification resulted in
capturing features larger than bars in the large-scale crests class. To distinguish between
bars and larger features, the standard deviation of each DEM cell in a moving window
size of 250x250 m is calculated. A suitable threshold to distinguish between bars and
larger features are 0.6 standard deviations. Finally, swash bars and linear bars are
distinguished by an area/perimeter-ratio, based on the assumption that linear bars has a
smaller ratio than swash bars, due to the different shapes. In this case, 4 were found to
be a suitable ratio threshold.

# 4 Results

## 4.1 Refraction correction and dead zone extent

The vertical adjustment of the LiDAR points due to refraction correction ($z_{\text{diff}}$) is linearly correlated with the water depth ($d$) (Fig. 6). An empirical formula was derived for this relationship and is given by the equation:

$$z_{\text{diff}} = 0.227 * d \text{ , } R^2 = 0.997 \tag{7}$$

A LiDAR point at 1 m water depth is vertically adjusted by approximately 0.23 m (Fig. 6). The variations around the linear trend in Fig. 6 are due to changing incidence angles of the laser beam that varies with the airplane attitude (roll, pitch and yaw).

The vertical extent of the dead zone is approx. 28 cm, determined by plotting the vertical difference between the shallowest and the deepest LiDAR point within 0.5 m cells − i.e. between the shallow surface and the deep surface (Fig. 7). The difference is manifested by an abrupt change at the dead zone, and the highest rate of change is shown to be at a water depth of approx. 28 cm.

## 4.2 Sub-decimetre accuracy and precision

The vertical root mean square error of the LiDAR data is ±4.1 cm, and the accuracy is ±8.1 cm with a 95% confidence level (Table 1 and Fig. 8A). The vertical precision of the LiDAR data with a 95 % confidence level is ±3.8 cm for the points on the frame, and ±7.6 cm for the points on the block (Table 1).

The horizontal accuracy calculated as the horizontal mean absolute error ($E_{\text{MA,xy}}$) is determined to ±10.4 cm, which is the average of the minimum distances between the four block corners and the edge of the block surface derived by the LiDAR data (Fig. 8B).

## 4.3 Point density and resolution

The average point density is 20 points per m$^2$, which equals an average point spacing of 20 cm (Table 2). The point density of the individual swaths varies between 7-13 points

per m$^2$, and the point density of the combined swaths in the study area, varies between
0-216 points per m$^2$, although above 50 points per m$^2$ are rare.

## 4.4  DEM and landforms

The altitudes in the studied section of the Knudedyb tidal inlet system range from -4
m DVR90 in the deepest parts of the flood channel and main channel to 21 m DVR90
on top of the beach dunes on Fanø (Fig. 9). Beach dunes and cottages of the village
Sønderho are clearly visible in the northern part of the study site (Fig. 9A-B). The
intertidal areas are generally flat, while the most varying morphology is found in the
area of the flood channel (Fig. 9C-D), and in the area close to the main channel (Fig.
9E-F). The flood channel is approximately 200 m wide in the western part and it divides
into two channels towards east. The bathymetry of the channel bed is clearly captured
by the LiDAR data in the eastern part, and also in the western part down to -4 m
DVR90, which approximately equals a water depth of 3 m at the time of survey. An
intertidal creek joins the flood channel from the north (Fig. 9D). From the flood channel
towards south, the tidal flat is vaguely upward sloping, until reaching two distinct swash
bars, which are rising 0.9 m above the surrounding tidal flat, reaching a maximum
altitude of 1.5 m DVR90 (Fig. 9E-F). Further south, the linear bars along the margin of
the main channel are clearly captured in the DEM (Fig. 9E).

## 4.5  Geomorphometric and morphological classifications

The geomorphometric and morphological classifications show that most of the study
site is located in the intertidal zone, and is mostly flat. That is manifested by the
dominating two classes; flats and intertidal flats (Fig. 10A-B). The geomorphometric
classification identifies slopes as stripes with NNW-SSE directionality across the flats.
These are following the direction of the survey lines, and thus, they are not real
morphological features but more an indication of lower precision of the LiDAR data,
especially at the outer beams of the swath. These swath artefacts are smoothed out in the
morphological classification by down-sampling the DEM to 2 m resolution, and
therefore, the intertidal flats appear uniform and seamless. The bar features close to the
main channel are well defined in the geomorphometric classification where they are
classified as large-scale crests and small-scale crests surrounded by slopes. In the

morphological classification, these are identified based on neighbourhood analyses and separated by the area/perimeter-ratio into two classes, swash bars and linear bars (Fig. 10C). Large-scale crests are also found on Fanø in the northern part of the area, and most of these are classified as beach dunes in the morphological classification. The geomorphometric classification identifies more large-scale crests along the banks of the flood channel, however, these are not actual bar features but they are identified as crests due to the nearby flood channel and creeks resulting in a positive broad scale BPI. In the morphological classification it is possible to distinguish between these "false" crests and actual bar features, by looking at altitude deviations at an even larger scale than the broad scale BPI. The intertidal creek in the NWern part of the area is a mix of depressions, slopes and small-scale crests in the geomorphometric classification, whereas it is relatively well defined and properly delineated in the morphological classification (Fig. 10D).

The geomorphometric classification identifies slopes along the banks of the main channel, flood channel and the intertidal creek, as well as in front of the beach dunes and along the edges of the swash bars and linear bars. The slopes seem particularly reliable at delineating the features in the intertidal zone; swash bars, linear bars and creeks. Depressions are primarily identified in the deepest detected parts of the main channel and in the flood channel, in the intertidal creek and in the beach dunes. Small-scale crests are found in the geomorphometric classification in locations which are high compared to its near surroundings. They are primarily seen as parts of the linear bars close to the main channel, in the beach dunes on Fanø and along the banks of the intertidal creeks.

A few small circular patches of approx. 5 m diameter with *Spartina Townsendii* (Common Cord Grass) located on the intertidal flat are classified as small-scale crests in the geomorphometric classification (Fig. 11). It clearly shows the capability of capturing relatively small features in the DEM and in the derived classification.

# 5 Discussion

## 5.1 Performance of the water surface detection method

The water surface in topobathymetric LiDAR surveys is most often detected from NIR LiDAR data, which is simultaneously collected along with the green LiDAR data (Collin et al., 2012; Guenther et al., 2000; Parker and Sinclair, 2012; Wang and Philpot, 2007). However, detecting the water surface based on the green LiDAR data provides a potential to perform topobathymetric surveys with just one sensor, thus optimizing the survey costs as well as data handling and storage.

The two critical issues risen by Guenther et al. (2000), as mentioned in the introduction, concerning the water surface detection with green LiDAR were thoroughly investigated in this study. The first issue, regarding the gap of detected water surface signals in the dead zone, is addressed by detecting the water surface based on areas which are known to be covered by water, and thereafter extending the water surface until it intersects the topography, so that also the dead zone is covered by the modelled water surface. The second issue, regarding uncertainty in the water surface altitude determination, is addressed using the results presented byMandlburger et al. (2013) who found a statistical relationship between the cloud of water surface points in the green LiDAR data and the water surface altitude derived from NIR LiDAR data. Mandlburger et al. (2013), however, did not describe the actual method of modelling the water surface, which is done in this study. Mandlburger et al. (2015), on the other hand, did propose a method for modelling the water surface, however, it was in a fluvial environment and the water level was based on manual determinations of cross sectional water levels. The water surface detection method in this study is thus new in combining the properties: 1) It is only using green LiDAR data, 2) it is based on automatic water level determination 3) it is applied in a tidal environment (can be applied in any coastal environment) and 4) it is open to the public and described in detail.

The developed water surface detection method is new but it must be pointed out that the assumption of a flat water surface leaves room for improvements for the future, especially if it is applied in a fluvial environment. Assuming a flat water surface is indeed a simplification of the real world, since the water surface in reality can be inclined, and it can be topped by waves.

## 5.2 Implications of the dead zone

The vertical extent of the dead zone is in this study determined to approx. 28 cm (Fig. 7), which means that no return signal is detected from the water surface when the water depth is less than 28 cm. As Guenther et al. (2000) explains, the dead zone poses a real challenge to the modelling of a water surface, because all submerged points, also those in less than 28 cm water depth, have to be corrected for refraction. With the water surface detection method proposed in this work this issue has been dealt with by extending the water surface into the dead zone, which makes it possible to correct the LiDAR points in 0-28 cm water depth for refraction. In this way, the implication of the dead zone along the channel edges is diminished, which is particularly beneficial in flat areas such as the Knudedyb tidal inlet system, where the dead zone may cover large areas depending on the tide (Fig. 12).

The implication of the dead zone along the channel edges is minimised, but the setting is different for the small ponds on the intertidal flats. They may have different water levels than in the large channels, but no detected water surface points, since the water depth in the ponds are generally less than the vertical extent of the dead zone, i.e. approx. 28 cm. The presented method is not capable of detecting a water surface in these ponds, which means that the bottom points of the ponds are not corrected for refraction. According to the calculated refraction (Fig. 6), omitting refraction correction of a 28 cm deep pond will result in -6 cm altitude error (naturally less error in shallower water).

## 5.3 Evaluation of the topobathymetric LiDAR data quality

The vertical accuracy of conventional topographic LiDAR has previously been determined to ±10-15 cm (Hladik and Alber, 2012; Jensen, 2009; Klemas, 2013; Mallet and Bretar, 2009). Only few previous studies have focused on the accuracy of shallow water topobathymetric LiDAR data (Mandlburger et al., 2015; Nayegandhi et al., 2009; Steinbacher et al., 2012). Nayegandhi et al. (2009) determined the vertical $E_{RMS}$ of LiDAR data in 0-2.5 m water depth to ±10-14 cm, which is above the ±4.1 cm $E_{RMS}$ found in this study (Table 1). Steinbacher et al. (2012) compared topobathymetric LiDAR data from a RIEGL VQ-820-G laser scanner with 70 ground-surveyed river cross sections, serving as reference, and found that the system's error range was ±5-10

cm, which is comparable to the ±8.1 cm accuracy found in this study. Mandlburger et al. (2015) compared ground-surveyed points from a river bed with the median of the four nearest 3D-neighbors in the LiDAR point cloud, and they found a standard deviation of 4.0 cm, which is almost equal to the ±4.1 cm standard deviation found in this study (Table 1). In comparison with these previous findings of LiDAR accuracy, the assessment of the vertical accuracy in this study indicates a good quality of the LiDAR data.

Mapping the full coverage of tidal environments, such as the Wadden Sea, require a combination of topobathymetric LiDAR to capture topography and shallow bathymetry and MBES to capture the deeper bathymetry. The two technologies make it possible to produce seamless coverage of entire tidal basins; however, merging the two products raises the question whether the quality of the data from the two different sources is comparable. Comparing the LiDAR accuracy with previous findings of accuracy derived from MBES systems indicates similar or slightly better accuracy from the MBES systems (Dix et al., 2012; Ernstsen et al., 2006a). Dix et al. (2012) determined the vertical accuracy of MBES data by testing the system on different objects and in different environments, and found the vertical $E_{RMS}$ to be ±4 cm. Furthermore, they tested a LiDAR system on the same objects and found a similar vertical $E_{RMS}$ of ±4 cm. The vertical $E_{RMS}$ of ±4.1 cm found in this study is very close to both the MBES accuracy and LiDAR accuracy determined by Dix et al. (2012). Another study by Ernstsen et al. (2006a) determined the vertical precision of a high-resolution shallow-water MBES system based on 7 measurements of a ship wreck from a single survey carried out in similar settings as the present study, namely in the main tidal channel in the tidal inlet just north of the inlet investigated in this study. They found the vertical precision to be ±2 cm, which is slightly better than the vertical precision of ±3.8 cm (frame) and ±7.6 cm (block) found in this study. Overall, accuracy and precision are within the scale of sub decimetres for both topobathymetric LiDAR and MBES systems, which enables the mapping of tidal basins with full coverage and with comparable quality.

Due to technical and logistical reasons, the data validation and the actual survey were carried out on different days and in different locations. Based on this, it is a fair question to ask, whether the determined quality actually represents the quality of the data within

the study site. In order to address this issue, the environmental conditions between the
two surveying dates, as well as the environmental differences, which may impact the
data quality, between the study site and the validation sites are compared.
The environmental conditions in the two surveying days were similar, with sunny
conditions, average wind velocities of 7-8 m/s (DMI, 2014) and significant wave
heights, measured west of Fanø at 15 m water, of approx. 0.5 m coming from NW
(DCA, 2014). However, the waves in the main channel, next to the study site, have been
observed in the 30 May LiDAR point cloud to be not more than 0.2-0.3 m, which can be
explained by the location of the study site in lee of the western most intertidal flats and
the ebb-tidal delta. The wave heights in the rest of the study area (flood channel and
intertidal ponds) were in the scale of sub decimetres. In comparison, there were no
waves at validation site 2 in Ribe Vesterå River during the 19 April LiDAR survey. As
already mentioned, the proposed water surface detection method has a shortcoming of
not modelling the waves, and this is a source of error in areas exposed to waves. The
precision of the seabed points within the study area are therefore expected to be worse
than the ±3.8 cm precision determined at validation site 2, because of the larger wave
exposure.
The water clarity/turbidity impacts the accuracy of the LiDAR data negatively, due to
scattering on particles in the water column, which causes the laser beam to spread
(Kunz et al., 1992; Niemeyer and Soergel, 2013). Moreover, part of the light is reflected
in the direction of the receiver, and such return signals can be difficult to distinguish
from the seabed return (Kunz et al., 1992). The turbidity was measured at validation site
2 and in the flood channel close to the study site during the 19 April survey by
collecting water samples and subsequently analysing the samples for suspended
sediment concentration (SSC) and organic matter content (OMC). The analyses showed
that the average SSC was higher in the flood channel (17.2 mg/kg) than in the river
(10.2 mg/kg). In contrast, the average OMC was lower in the flood channel (25.5 %)
than in the river (40.0 %). These observations indicate that 1) the underwater precision
is assessed in a location with higher turbidity than the environment within the study site;
therefore, the turbidity cannot be a cause of lower precision in the study site, and 2) the
penetration depth seems to be controlled by the OMC rather than by the SSC. This is
new knowledge, since no previous studies (from what we know) have investigated the
relative effect of organic matter as opposed to inorganic matter on the laser beam
penetration depth. However, in order to determine the relationship with statistical
confidence, a more comprehensive study is needed, involving measurements of
penetration depth at different SSCs and OMCs, and without disturbance from other
environmental parameters.

## 5.4 Spatial variations of topobathymetric LiDAR data quality

The quality of spatial datasets is often provided as single values, such as ±8.1 cm for the
vertical accuracy in this case, and then the determined value represents the
accuracy/precision of the whole dataset. However, in reality the value is only a measure
of the local quality at the location where the assessment is conducted. The quality of the
dataset varies spatially, and one way to illustrate that is to extract the maximum vertical
difference between the LiDAR points of the processed point cloud within every 0.5×0.5
m cell throughout the study site (Fig. 13). In flat areas, without multiple return signals,
this shows the spatially varying precision of the dataset. There are large differences on
Fanø, which is expected due to vegetation causing multiple LiDAR returns from both
the vegetation canopy and from the bare ground. In contrast, the differences on the very
gently sloping, non-vegetated tidal flat are up to 10 cm, and there is no simple and
natural reason for that variation. A range of factors contribute to the observed
variations:
Laser beam incidence angle: The incidence angle, at which the laser beam hits the
ground/seabed, is determined by a combination of the scan angle, the water surface
angle and the terrain slope. The shape of the footprint is stretched with larger incidence
angles, and this effect can cause pulse timing errors in the detected signal, which leads
to a decreasing vertical accuracy (Baltsavias, 1999). The error associated with larger
scan angles is generally causing the outer beams, toward the swath edges, to attain a
lower accuracy (Guenther, 2007). This is a reason for the observed variations along the
swath edges (Fig. 13). Terrain slopes has the same effect of decreasing the vertical
accuracy due to the footprint stretching. The measured altitude tend to be biased toward
the shallowest point of the slope within the laser beam (Guenther, 2007). However, the
influence of slope is not crucial in the Knudedyb tidal inlet system, since it is generally
a very flat area.
Vertical bias between overlapping swaths: Areas covered by more than a single swath
tend to show more vertical variation in the LiDAR point measurements. This can be
caused by variance/error in the GPS measurements and/or IMU errors (Huising and
Gomes Pereira, 1998). The vertical bias between swaths has been observed in the point
cloud to be up to 5 cm, but it is varying throughout the study site. In most environments,
a bias of 5 cm would be unnoticeable, but because of the large and very flat parts of the
Knudedyb tidal inlet system, even a small bias becomes readily evident.
Water depth: The accuracy and precision are expected to be lower as the laser beam
penetrates deeper into the water column (Kunz et al., 1992). The laser beam footprint is
diverging as it moves through the water column, resulting in a larger footprint on the
seabed. The altitude of the detected point is thus derived from the measurement on a
larger area on the seabed, which will decrease the vertical accuracy, as well as decrease
the capability of detecting small objects. With this in mind, the lower precision at the
frame compared to the block is opposite of what would be expected, since the frame is
below water and the block is on land. In this case, other factors, such as overlapping
swaths and/or scan angle deviations, have more influence on the precision than the
water depth. Also, it should be remembered that the frame surface was close to the
water surface, and the effect of the water depth on the precision would most likely be
more evident if it was located in deeper water.
Additional factors, beside the ones mentioned above, may influence the quality of
LiDAR datasets. For instance, a dense vegetation cover of the seabed or breaking waves
that makes the laser detection of the seabed almost impossible. However, these factors
do not have a great influence in the studied part of the Knudedyb tidal inlet system, and
thus they are not further elaborated.

## 25    **5.5   Evaluation of the morphological classification**

The morphological classification presented in this study is based on the studied section
of the Knudedyb tidal inlet system. The overall concept of using tidal range, slope and
variations of the altitude at different spatial scales proves to be a reliable method for
delineating the morphological features in this tidal environment. The concept, however,
can be applied in other environments. The specific thresholds in the classification
determined in this study may deviate in other areas. Morphological features of different

sizes require steps of other spatial scales in the neighbourhood analyses to produce a successful classification. In the future the classification method will be improved by implementing an objective method for determining the scales, which can make it applicable in areas with different morphological characteristics. Such an objective scale determination method is presented by Ismail et al. (2015), who determined the scales based on the variance of the DEM at progressively larger window sizes. In this way, the sizes of the morphological features are determining the scales for the classification.

## 5.6   Using topobathymetric LiDAR data to map morphology in a highly dynamic tidal environment

The study demonstrates the capability of green topobathymetric LiDAR to resolve fine-scale features, while covering a broad-scale tidal inlet system. Collecting topobathymetric LiDAR data with a high point density of 20 points/m$^2$ on average enables detailed seamless mapping of large tidal environments, and the LiDAR data has further proved to maintain a high accuracy. The combined characteristics of mapping with high resolution and high accuracy in a traditionally challenging environment provide many potential applications, such as mapping for purposes of spatial planning and management, safety of navigation, nature conservation, or morphological classification, as demonstrated in this study. The developed LiDAR data processing method is tailored to a morphological analysis application. The best representation of the morphology is mapped by gridding the average value of the LiDAR points into a DEM with a $0.5 \times 0.5$ resolution. Other applications would require different gridding techniques. For instance hydrographers, who are generally interested in mapping for navigational safety, would use the shallowest point for gridding. However, the overall method for processing the point cloud can be used regardless of the application. Only the last and least challenging/time consuming step of gridding the point cloud into a DEM may vary depending on the application.

Applying topobathymetric LiDAR data for morphological analyses in tidal environments enables a holistic approach of seamlessly merging marine and terrestrial morphologies in a single dataset. In order to map the morphology of tidal environments in full coverage, however, a combination of topobathymetric LiDAR and MBES swath data is required. The comparable quality and resolution of LiDAR and MBES data gives

a potential to map large scale tidal environments, such as the Wadden Sea, in full
coverage and with high resolution and high accuracy.

## 6   Conclusions

A new method was developed for processing raw topobathymetric LiDAR data into a
digital elevation model with seamless coverage across the land-water transition zone.
Specifically a procedure was developed for water surface detection utilizing automatic
water level determination from only green LiDAR data in a tidal environment. The
method relies on basic principles, and in general the entire processing method is
described with a high level of detail, which makes it easy to implement for future
studies. The water surface detection method presented in this work did not take into
account the variation in wave heights and surface slopes, which therefore constitutes a
challenge to be addressed in future studies.
The vertical accuracy of the LiDAR data was determined by object detection of a
cement block on land to ±8.1 cm with a 95% confidence level. The vertical precision
was determined at the cement block to ±7.6 cm, and ±3.8 cm at a steel frame, placed
just below the water surface. The horizontal mean error was determined at the block to
±10.4 cm. Overall, vertical and horizontal precision is within sub decimetre scale.
A seamless topobathymetric digital elevation model was created in a 4×0.85 km section
in the Knudedyb tidal inlet system. An average point density of 20 points per m$^2$ made it
possible to create an elevation model of 0.5×0.5 m resolution without significant
interpolation. The model extends down to water depths of 3 m, which was the maximum
penetration depth of the laser scanning system at the given environmental conditions.
Measurements of suspended sediment concentration and organic matter content indicate
that the penetration depth is limited by the amount of organic matter rather than the
amount of suspended sediment.
The vertical "dead zone" of the LiDAR data has been determined to be approx. 0-28 cm
in the very shallow water.
A morphological classification method was developed for classifying the area into 6
morphological classes: swash bars, linear bars, beach dunes, intertidal flats, intertidal

creeks and subtidal channels. The morphological classification method is based on parameters of tidal range, terrain slope, a combination of various statistical neighbourhood analyses with varying window sizes and the area/perimeter-ratio of morphological features. The concept can be applied in any coastal environment with knowledge of the tidal range and the input of a digital elevation model; however, the thresholds may need adaptation, since they have been determined for the given study area. In the future the classification method should be improved by implementing an objective method for determining thresholds, which makes it immediately applicable across different environments.

Overall this study has demonstrated that airborne topobathymetric LiDAR is capable of seamless mapping across land-water transition zones even in environmentally challenging coastal environments with high water column turbidity and continuously varying water levels due to tides. Furthermore, we have demonstrated the potential of topobathymetric LiDAR in combination with morphometric analyses for classification of morphological features present in coastal land-water transition zones.

## Acknowledgements

This work was funded by the Danish Council for Independent Research | Natural Sciences through the project "Process-based understanding and prediction of morphodynamics in a natural coastal system in response to climate change" (Steno Grant no. 10-081102) and by the Geocenter Denmark through the project "Closing the gap! – Coherent land-water environmental mapping (LAWA)" (Grant no. 4-2015).

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

Table 1: Vertical accuracy and precision of the LiDAR point measurements, in terms of
minimum error ($E_{min}$), maximum error $(E_{max})$, standard deviation ($\sigma$), mean absolute
error ($E_{MA}$), root mean square error ($E_{RMS}$) and the 95% confidence level ($Cl_{95\%}$).

| Accuracy/ Precision | Object | Best-fit plane | # points n | $E_{min}$ (cm) | $E_{max}$ (cm) | $\sigma$ (cm) | $E_{MA}$ (cm) | $E_{RMS}$ (cm) | $Cl_{95\%}$ (cm) |
|---|---|---|---|---|---|---|---|---|---|
| Accuracy | Cement block | GCPs | 227 | 0.01 | 12.1 | 4.1 | 3.5 | ±4.1 | ±8.1 |
| Precision | Cement block | Point cloud | 227 | 0.04 | 12.9 | 3.9 | 2.8 | ±3.9 | ±7.6 |
| Precision | Steel frame | Point cloud | 46 | 0.02 | 5.5 | 2.0 | 1.6 | ±1.9 | ±3.8 |

1 Table 2: LiDAR point spacing and density for all the 11 individual swaths, which

2 covered the study area, and for the combined swaths.

| Swath number | 1 | 2 | 3 | 4 | 5 | 6 | 7 | 8 | 9 | 10 | 11 | All |
|---|---|---|---|---|---|---|---|---|---|---|---|---|
| Point spacing (m) | 0.30 | 0.30 | 0.36 | 0.31 | 0.36 | 0.32 | 0.37 | 0.29 | 0.35 | 0.36 | 0.28 | 0.20 |
| Point density (pt./m$^2$) | 10.8 | 10.8 | 7.8 | 10.2 | 7.5 | 9.6 | 7.2 | 11.7 | 8.0 | 7.8 | 12.7 | 19.6 |

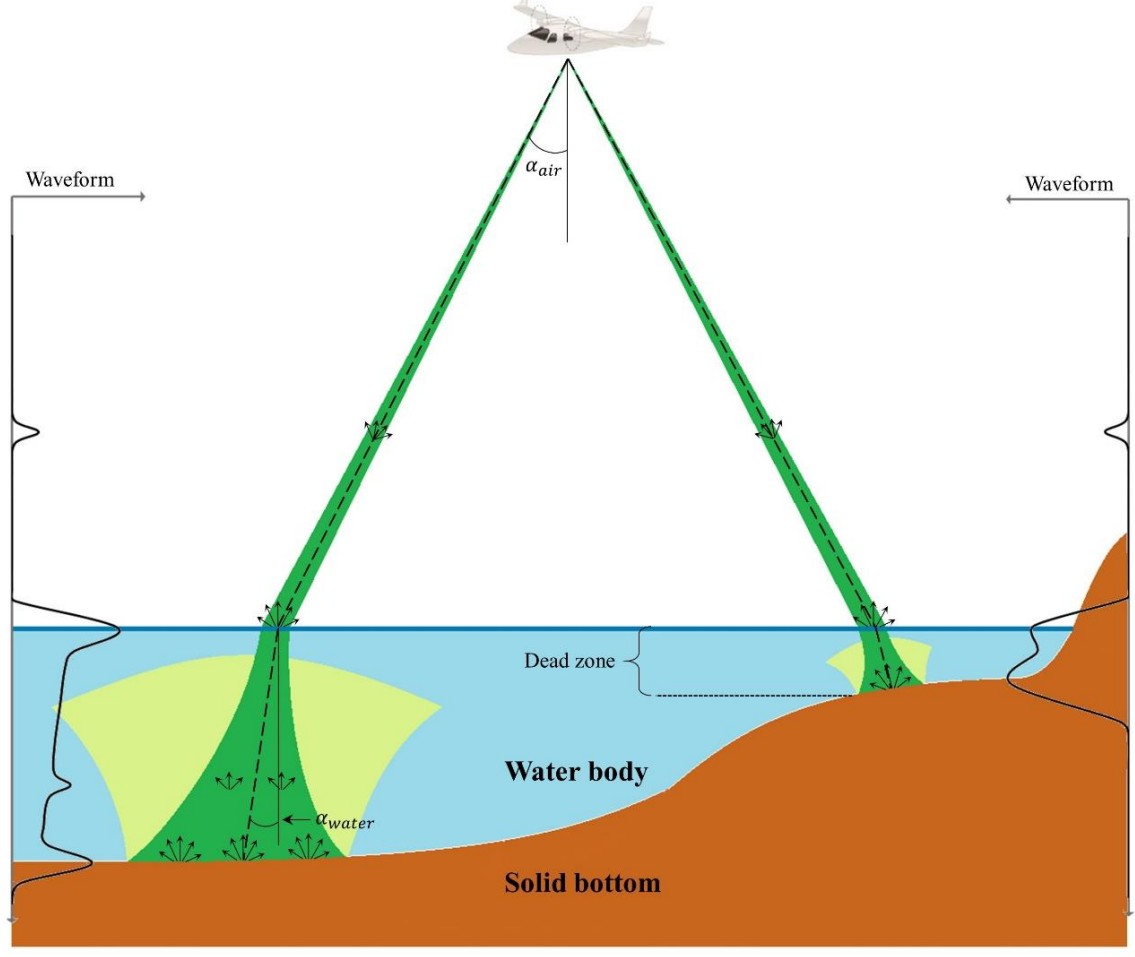

Figure 1: Conceptual sketch of the laser beam propagation and return signals. The beam
refracts upon entering the water body, and it diverges as it propagates through the water
column. Return signals are produced both in the air, at the water surface, in the water
column and at the seabed. The LiDAR instrument has limited capability in very shallow
water (the "dead zone" in the figure) because the successive peaks from the water
surface and the seabed are not individually separated in time and amplitude. Only the
largest peak, which is from the seabed, is detected.

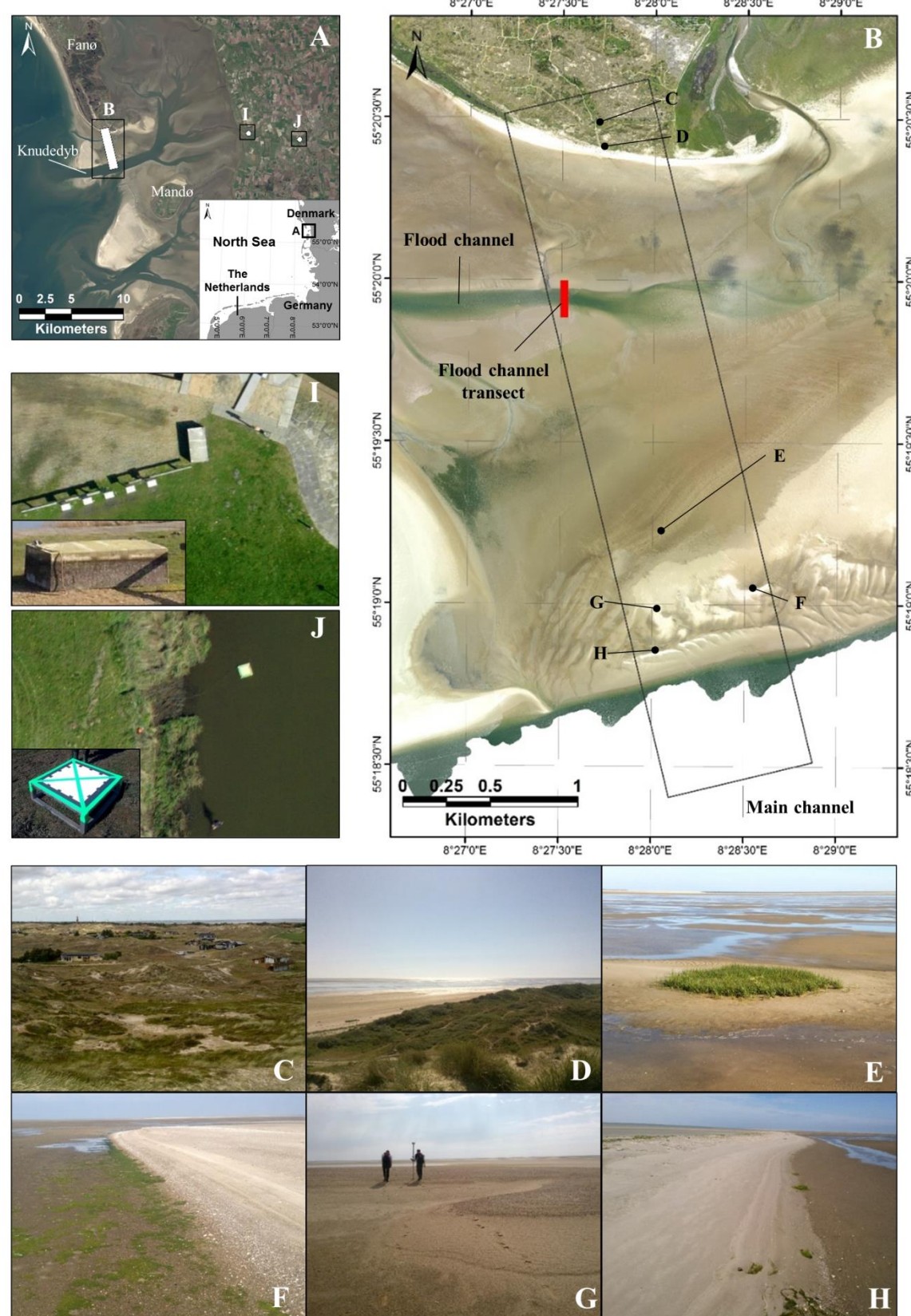

Figure 2: A) Overview of the study area location in the Danish Wadden Sea and the
specific locations of the study site (B) and the two validation sites (I and J) (22 April
2015 satellite image, Landsat 8). B) The study site in the Knudedyb tidal inlet system
(30 May 2015 Orthophoto, AHM). C) Cottages in the dunes on Fanø. D) Beach dunes
on Fanø. E) Patch of *Spartina Townsendii* (Common Cord Grass). F-G) Swash bars. H)
Linear bar. I) Validation site 1 with a cement block on land, used for accuracy and
precision assessment (19 April 2015 orthophoto, AHM). J) Validation site 2 with a steel
frame in Ribe Vesterå River, used for precision assessment (19 April 2015 orthophoto,
AHM).

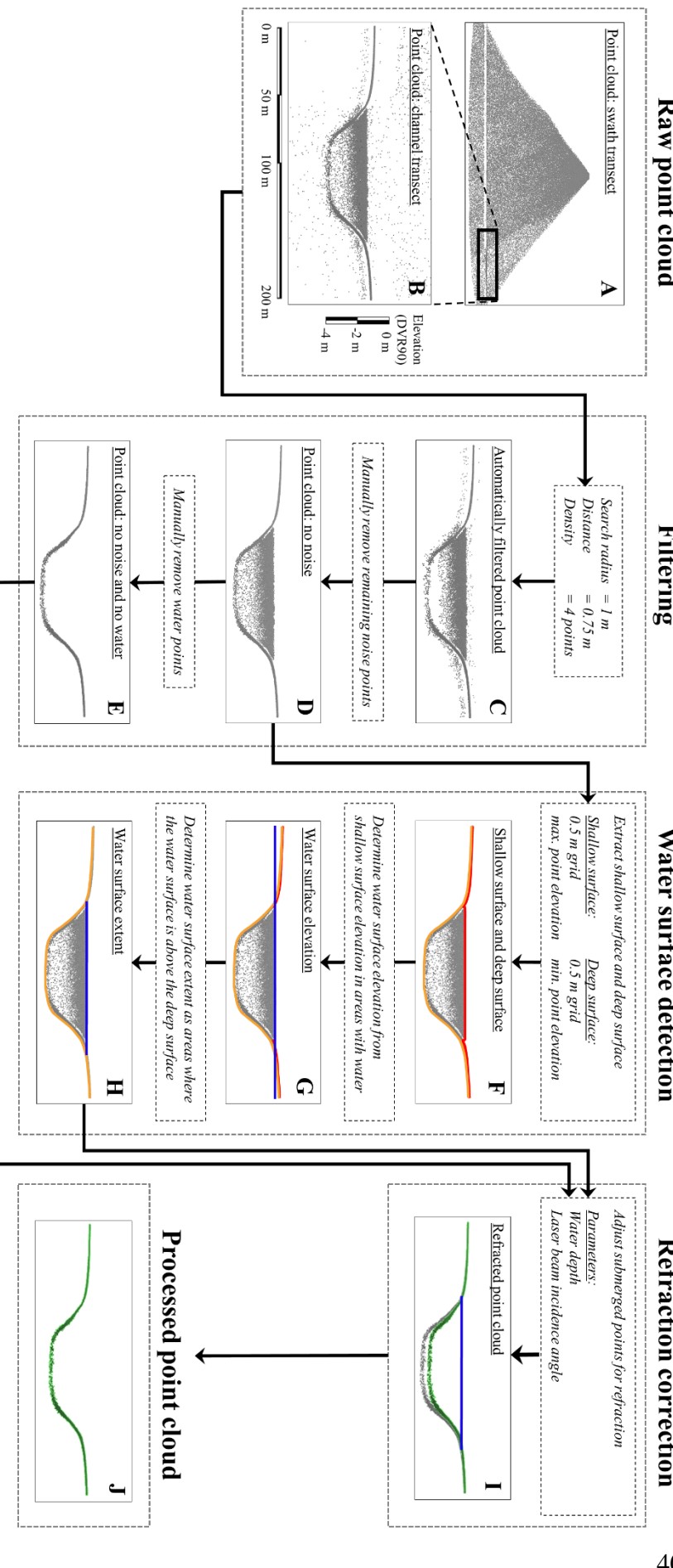

Figure 3: Workflow for processing the LiDAR point cloud. A) Point cloud from a single swath with points ranging from -100 m to 300 m elevation. B) Zoom-in on a cross section of the flood channel with altitudes exaggerated ×15 for visualization purpose. C-E) Method for filtering the point cloud. F-H) Method for detecting a water surface (blue) based on the extraction of a shallow surface (red) and a deep surface (orange). I) Correction for the effect of refraction on all the submerged points. J) Processed point cloud

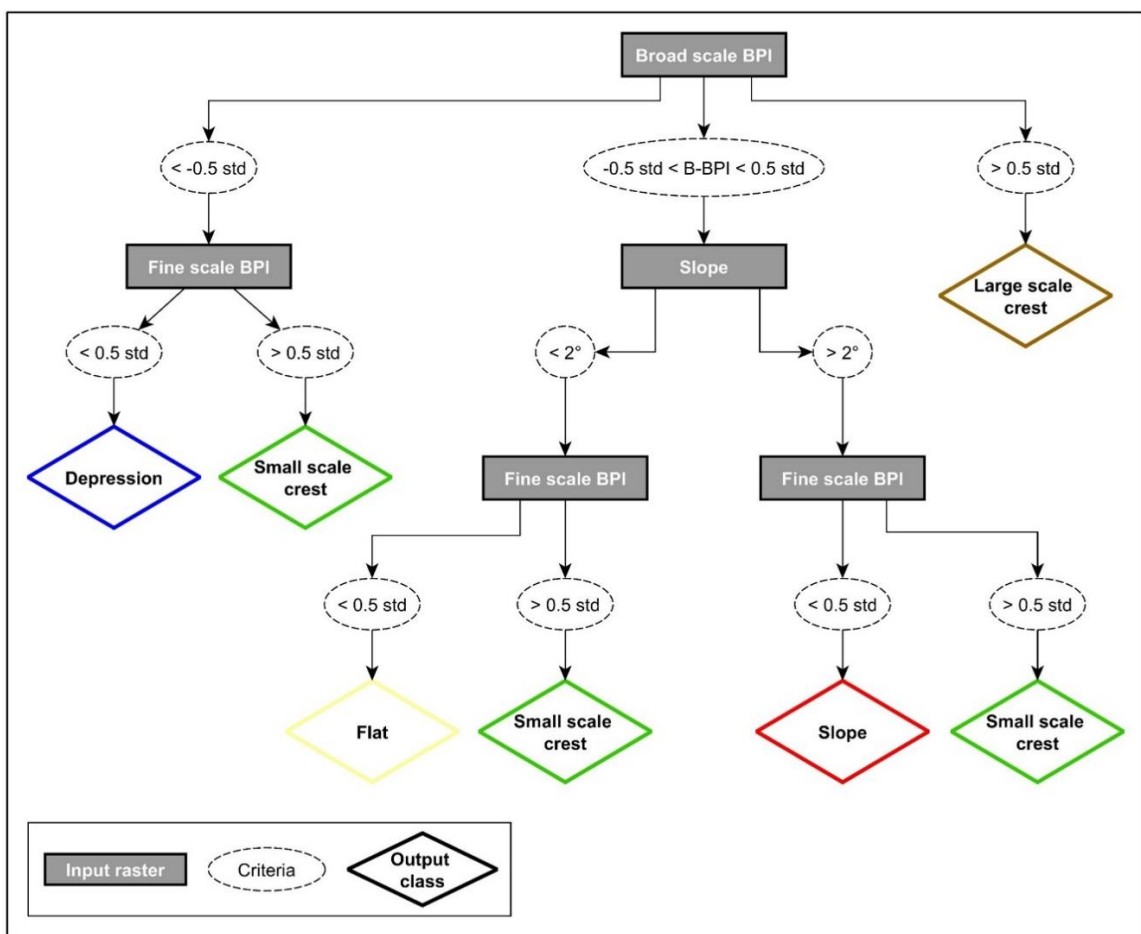

2 Figure 4: Classification decision tree, showing how the geomorphometric classification

3 was conducted in the Benthic Terrain Model tool.

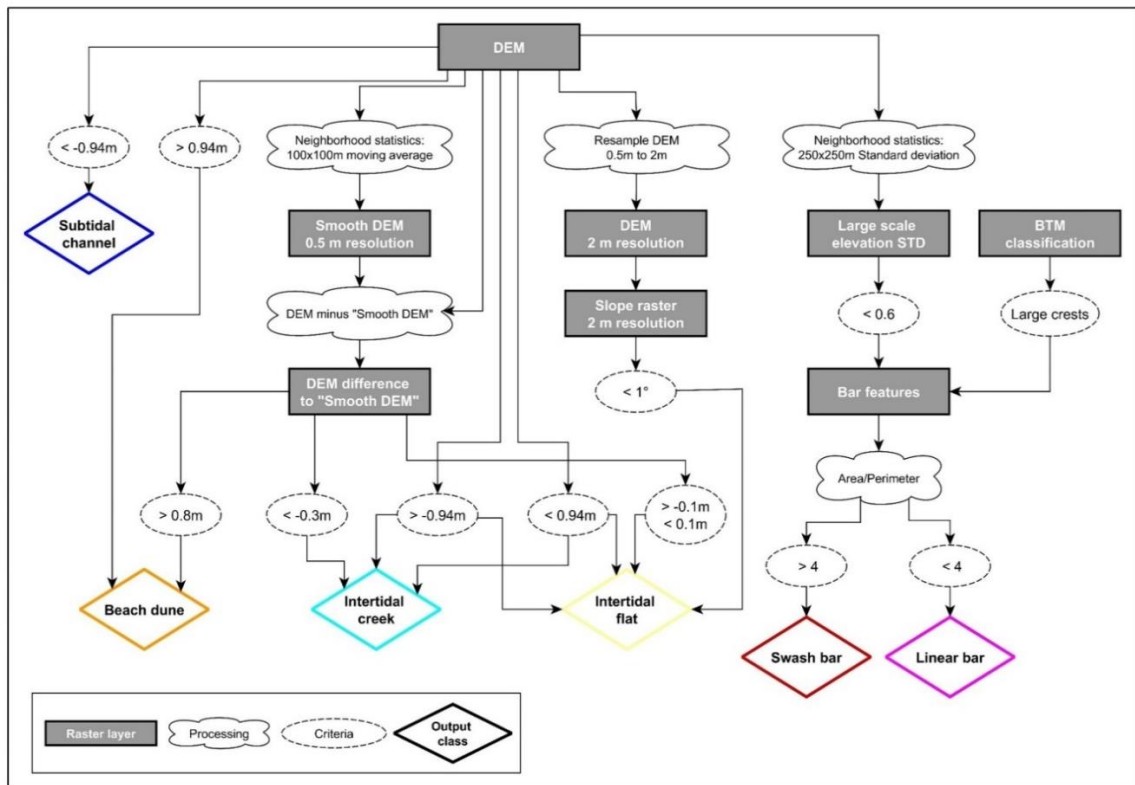

2    Figure 5: Classification decision tree of the morphological classification. All steps were

3    performed in ArcGIS.

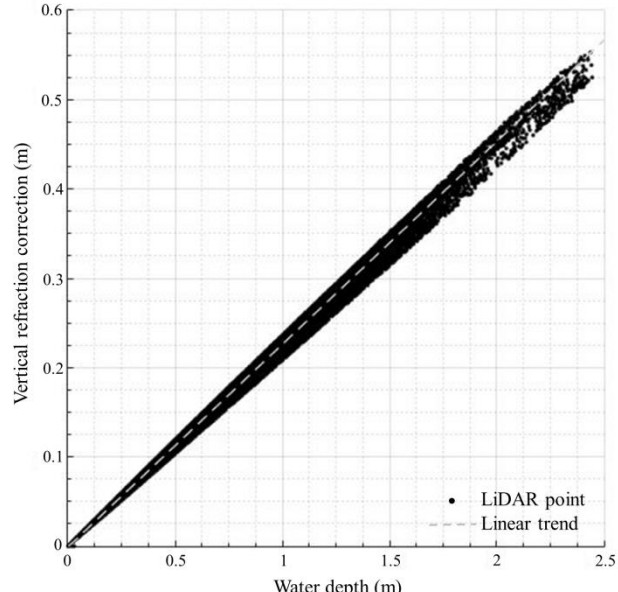

3    Figure 6: Vertical adjustment of the refracted LiDAR points from the flood channel

4    transect            (see          location        in        Fig.        2C).

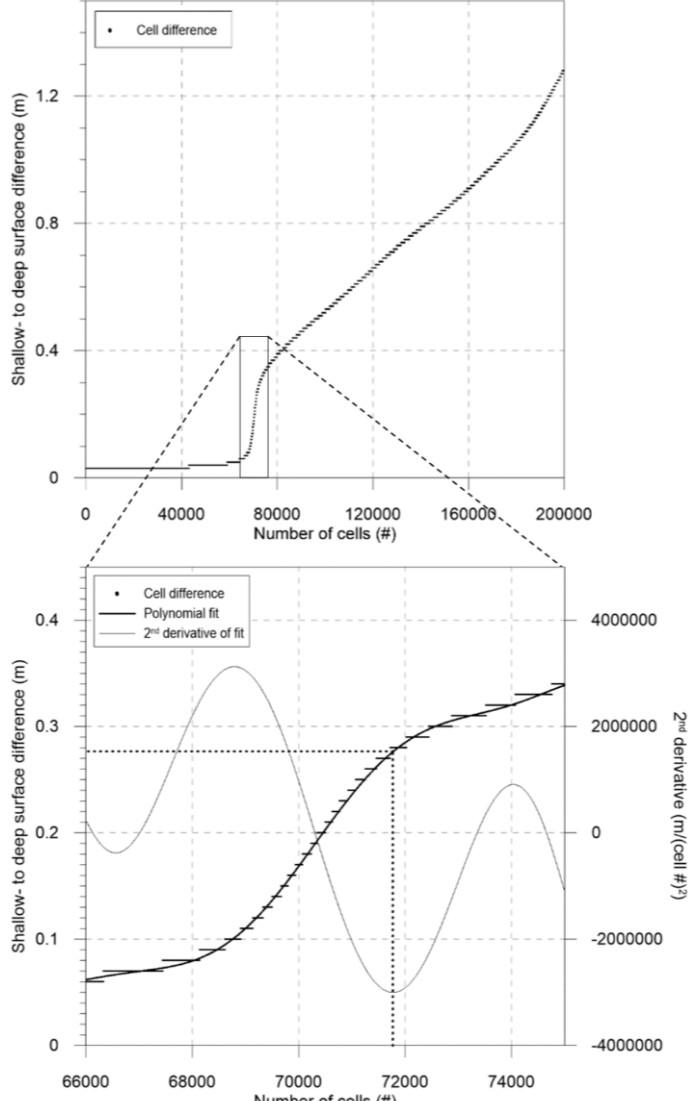

Figure 7: Vertical difference between the shallowest and the deepest LiDAR point
within 0.5 m grid cells in the land-water transition zone.  The abrupt change is caused
by the dead zone. The vertical extent of the dead zone is determined to approx. 28 cm,
derived by the maximum rate of change of a polynomial fit through the points.

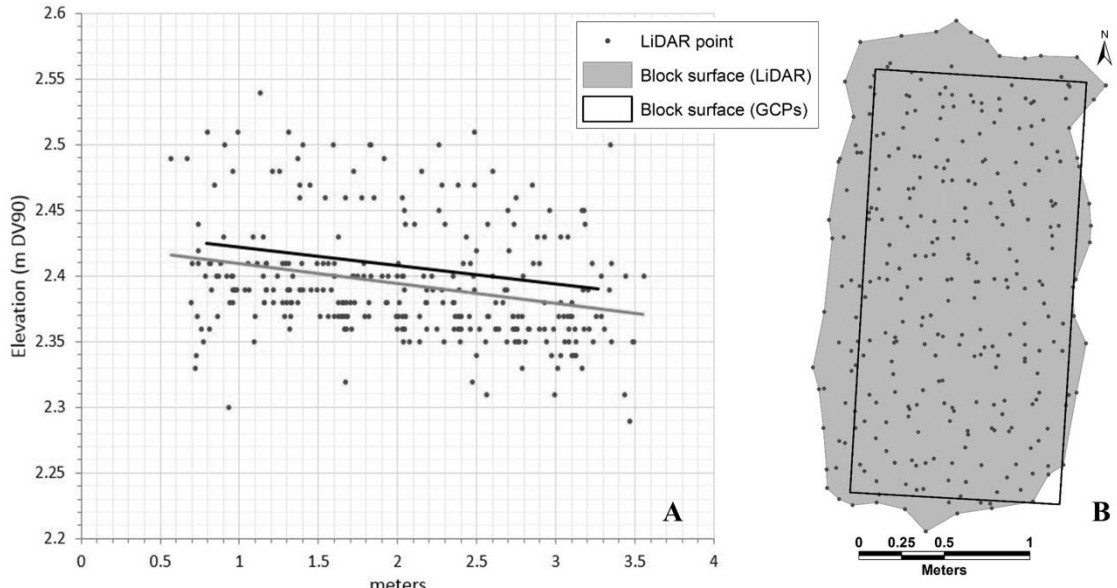

Figure 8: Vertical and horizontal distribution of the LiDAR points describing the block
surface and the block surface derived from Ground Control Points (GCPs). A) LiDAR
points (grey dots) compared to the GCP block surface (black line) for determining the
vertical accuracy. The grey line shows the LiDAR block surface as a best-linear-fit
through the points. B) Block surface derived from the four GCP corner points and the
block surface derived by the perimeter of the LiDAR points.

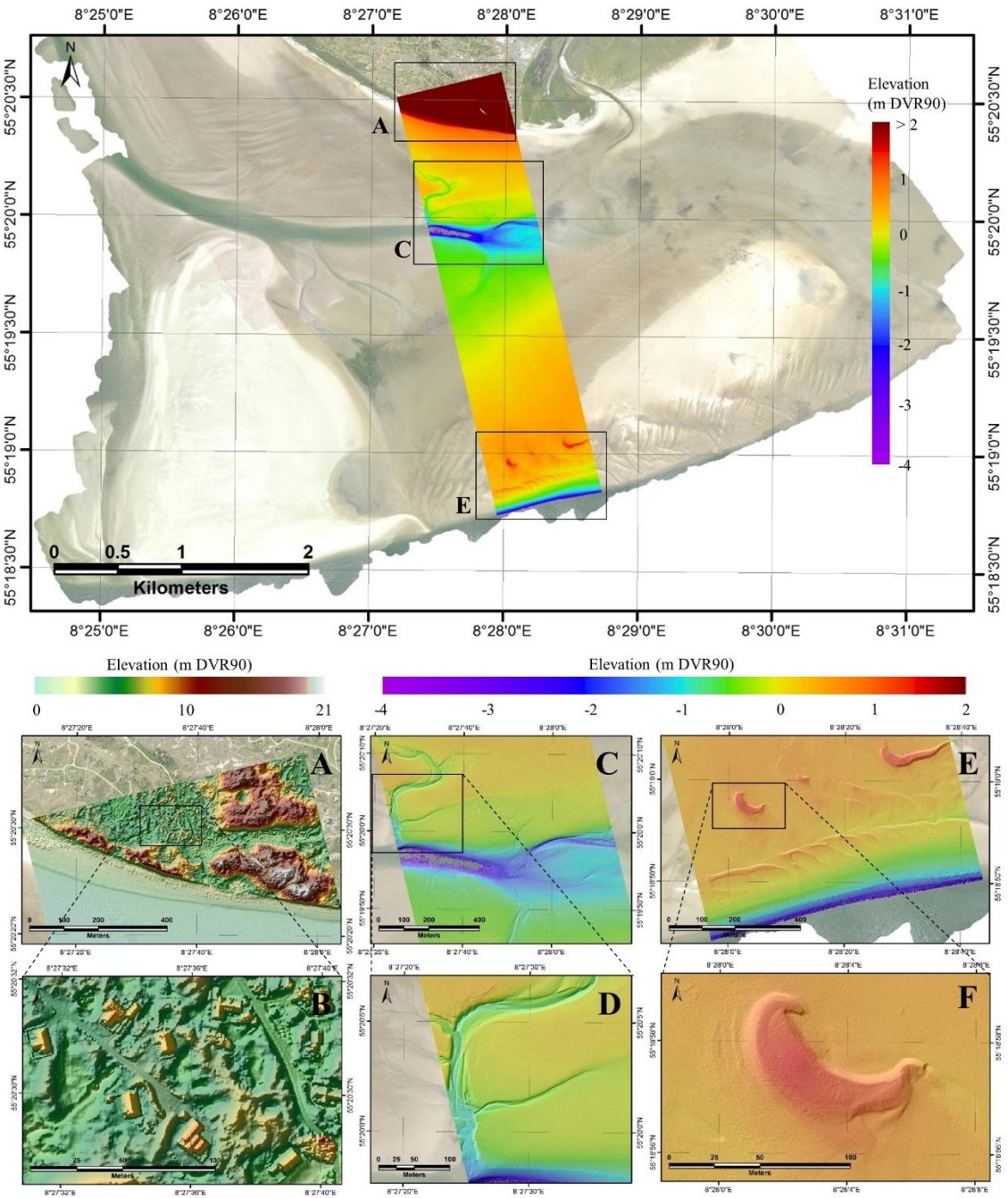

Figure 9: Topobathymetric DEM across the northern part of the Knudedyb tidal inlet system with close-up views of different detail level on specific areas. A hill shade is draped upon the close-ups for improved visualization of morphological features. A) Northern section with beach dunes and cottages. B) Cottages. C) Mid-section with the flood channel. D) Closer view on an intertidal creek. E) Southern section with swash bars, linear bars and bathymetry of the main channel. F) Swash bar.

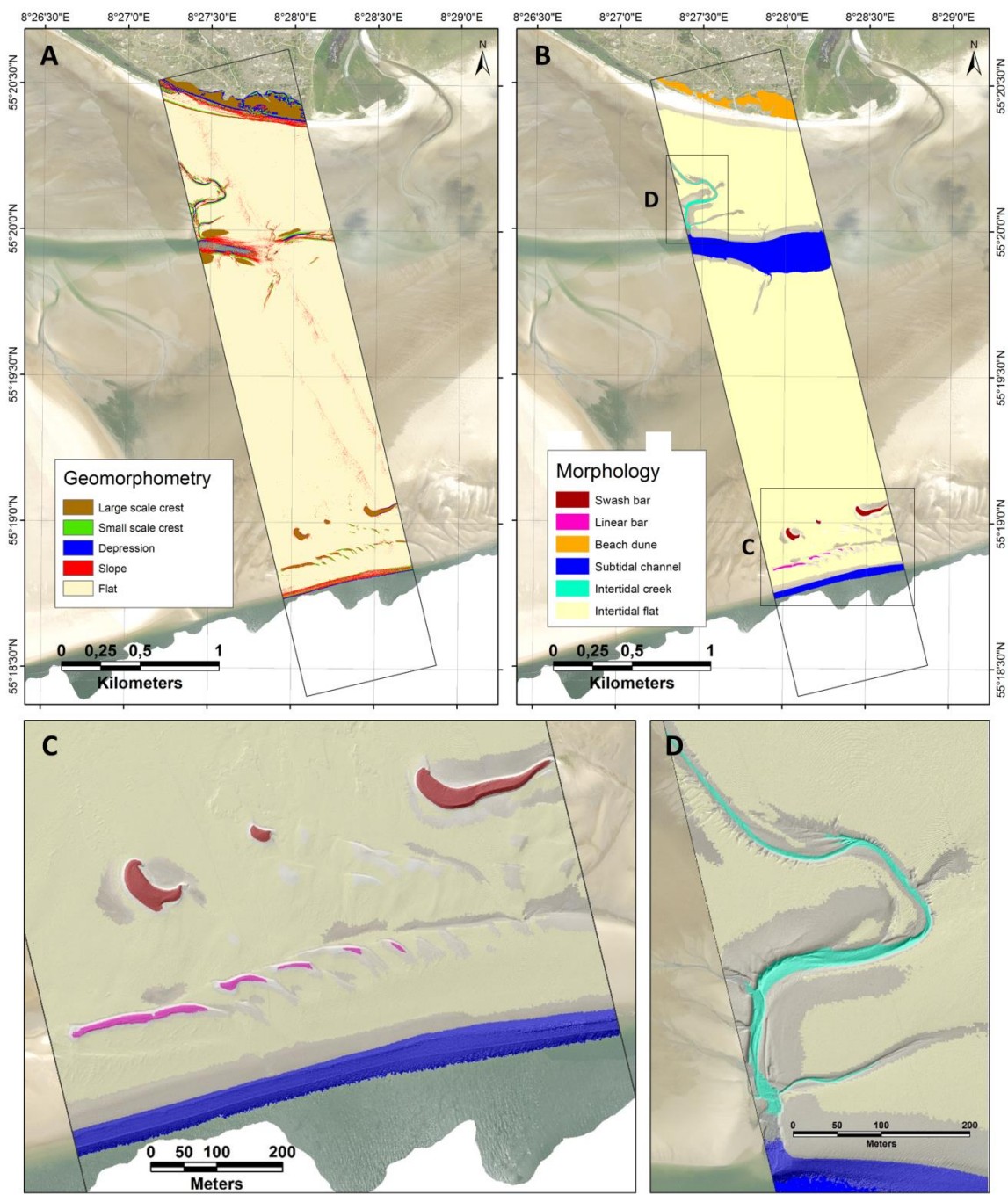

Figure 10: Two classifications of the investigated section in Knudedyb tidal inlet
system, derived from a topobathymetric DEM. A) Geomorphometric classification. B)
morphological classification. C) Zoom-in on the intertidal creek in the morphological
classification. D) Zoom-in on the swash bars and linear bars close to the main channel
in the morphological classification. A hillshade of the DEM is draped over C and D.

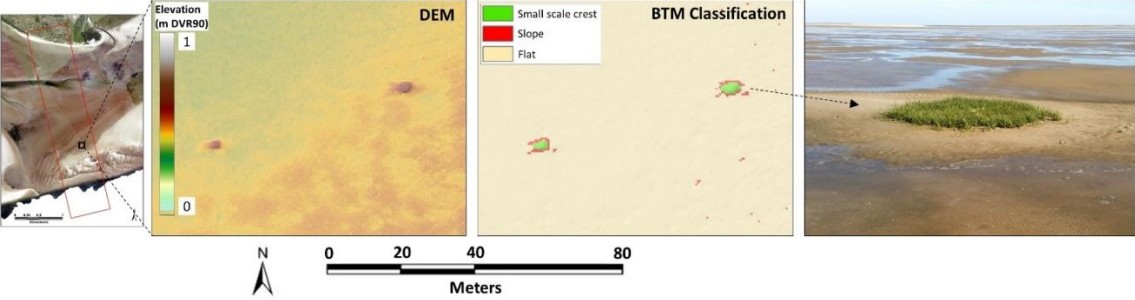

2 Figure 11: Vegetated mounds on the intertidal flat are clearly visible in the DEM and

3 classified as small-scale crests in the geomorphometric BTM classification. To the right

4 is an image of one of the patches.

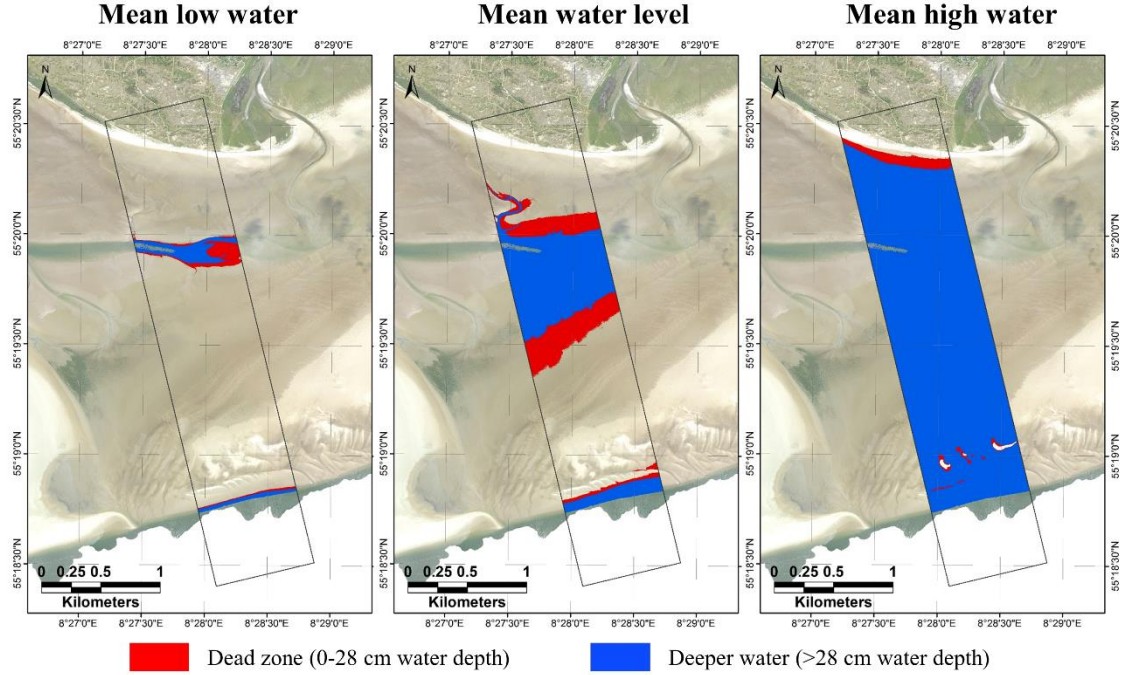

2 Figure 12: Horizontal extent of the dead zone in the studied area at mean low water,

3 mean water level and mean high water.

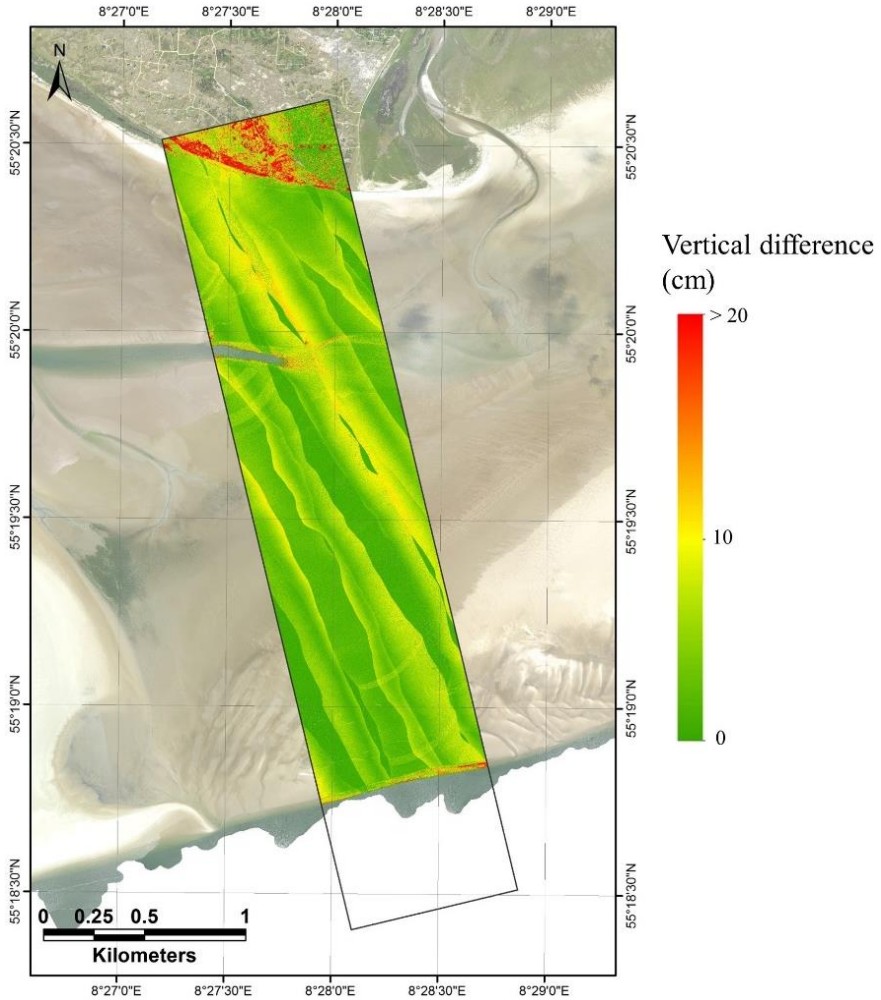

Figure 13: Vertical difference between the highest and the lowest LiDAR point within 0.5×0.5 m grid cells.