# Peer review of "Processing and performance of topobathymetric LiDAR"

_Hydrology and Earth System Sciences, 2016_

## Referee Comment (RC1) · Anonymous Referee #1 · 16 Feb 2016

General comments:

This manuscript describes a workflow to process a raw bathymetric LiDAR dataset to generate a digital terrain model of a tidal inlet system in the Danish Wadden Sea. The workflow allows extracting both the surface of the water and the seabed from the point cloud. As it stands, this paper reads more like a technical report. My understanding is that the method used to extract the water surface is novel, but I found difficult to grasp the novelty while reading the paper. Despite having valid scientific approach and applied methods, the work is not much grounded in the literature. Many key papers are not mentioned and the published works in water surface detection are not discussed, which prevents effectively identifying how this paper fills a gap in the knowledge. I also believe that the paper lacks context: it is unclear why the dataset was collected. This can have important implications for its processing. For instance, the workflow could not

be used to process a dataset collected to identify hazards to coastal navigation; hydrographers standards and protocols always identify the shallowest point of the seabed and use the worst case scenario in terms of low water level, while the workflow introduced in this paper uses the deepest point of the seabed and the maximum point of the water level. The discussion is quite short and could be more complete if wider scientific implications would be discussed. Also, I understand that the study of geomorphometry includes issues of data collection, but I find that this manuscript lacks some examples of morphological quantitative measurements that would make it more suitable for publication in a special issue on geomorphometry. Finally, I think that this paper has potentially many candidates for "lessons learned" that could help plan future surveys, but these take-home messages are not made explicit in the manuscript.

Here is my evaluation of the paper based on HESS review guidelines:

1. Does the paper address relevant scientific questions within the scope of HESS?

This paper does address relevant questions within the scope of HESS. However, the questions addressed are more technical than scientific. I think that the paper would benefit from being put in context within an application that would show how the proposed workflow can actually help answer scientific questions relevant to hydrology and earth systems.

2. Does the paper present novel concepts, ideas, tools, or data?

It is difficult to grasp the novelty in this paper. The essence of the workflow that is presented (i.e. raw data, correction, automatic and manual filtering, detection) is very similar to what is implemented in many bathymetric or LiDAR data processing software (e.g. Caris, Fledermaus, etc.). On page 4 (lines 6-8), the authors state that "The overall processing steps are known, but there is no standard or universal approach for dealing with the individual steps. In particular, there is no definitive method for detecting a water surface from topobathymetric LiDAR data." Methods however exist to detect a water surface from LiDAR data. For instance, water surface detection is

commonly performed by combining a laser in the green wavelengths with a laser in the near-infrared wavelengths, or if limited to only one laser, by identifying the different returns in the waveform. The novelty would be clearer and the paper would much benefit from a discussion on the state-of-the-art methods of water surface detection (i.e. a more complete literature review), and a clear argument on how the proposed method is different and/or better than the existing ones.

3. Are substantial conclusions reached?

The authors demonstrated the ability to use bathymetric LiDAR to detect fine-scale features in the coastal environment, which in itself has been demonstrated before in other studies. As mentioned in the previous points, the conclusions could be more substantial if the dataset was put in a more applied context (e.g. within a geomor-phometric application to study morphodynamics in the tidal inlet system), and if it was made clearer how the proposed method improves water surface detection compared to existing methods. Why should we adopt this method over another one? The authors need to convince us that their method is better in some ways (e.g. cost-efficiency, accuracy, extraction of relevant information, potential applications?).

4. Are the scientific methods and assumptions valid and clearly outlined?

Except for a few minor issues (see specific comments below), the scientific methods are valid and clearly outlined. I particularly appreciate that the authors detailed the assumptions that were made and discussed their implications.

5. Are the results sufficient to support the interpretations and conclusions?

Since it is a very technical paper, it does not have much interpretation. However, the results support the conclusions.

6. Is the description of experiments and calculations sufficiently complete and precise to allow their reproduction by fellow scientists (traceability of results)?

The authors did well in describing the processing and calculations which would allow

replicability.

7. Do the authors give proper credit to related work and clearly indicate their own new/original contribution?

As previously mentioned, the literature review is very general, mainly focused on the use of LiDAR. It does not go very deep in the methods that were tested or are commonly used to process the data, which prevents the clear identification of the authors' own new/original contribution.

8. Does the title clearly reflect the contents of the paper?

The title could be more representative. I suggest adding a word referring to "workflow" or "protocol". Also, the authors need to make sure that the system used was really "topobathymetric" (see specific comment below). Finally, I wonder why the authors mention accuracy in the title but not precision, which they measure. I suggest using a more general term such as "quality".

9. Does the abstract provide a concise and complete summary? The abstract is concise but not complete. There is no mention of the goal of the paper, no mention of water surface detection, and no mention of the general steps of the methods. It goes directly from the general mention of what was done ("a method is developed to...") to the results.

10. Is the overall presentation well structured and clear?

Overall, this paper is clear, concise and well-structured. I really enjoyed the quality of figures and tables.

11. Is the language fluent and precise?

The use of English language would need a bit of work, but nothing of big concern.

12. Are mathematical formulae, symbols, abbreviations, and units correctly defined and used?

Yes, everything was well defined. However, some of the equations presented in the introduction may not be necessary as they were not used directly by the authors in the methods. On the other hand, I understand that they may help readers with limited knowledge in LiDAR to understand the related concepts.

13. Should any parts of the paper (text, formulae, figures, tables) be clarified, reduced, combined, or eliminated?

I think that more background should be given on the scientific literature that looked at extracting the water surface from LiDAR. Also, the discussion could be improved by discussing the implications of the results for the wider scientific community. For instance, why not using multi-spectral LiDAR systems, or wavelengths that penetrate deeper? The system could not survey deeper than 3 m, a depth that is usually not covered by multibeam echosounders. Based on that and the fact that a gap in data would appear if merging with other datasets, can the authors really claim that their data provide a seamless coverage of the land-water transition zone?

14. Are the number and quality of references appropriate?

While reading the paper for the first time, I had the impression that many claims would benefit from a reference. I counted only 25 references in the list, which includes two technical reports, one thesis, one magazine, three user manuals, one textbook, and two abstracts from conference proceedings. Two of the remaining references are used to characterise the study area (i.e. tidal prism, average depth and width of the channels). I believe that the authors need more than 13 peer-reviewed full-length articles to set their work into the relevant literature and context.

Based on these 14 points, I do not recommend the manuscript for publication in HESS in its current condition. However, I laid down some specific comments below in the hope of helping the authors to improve their paper.

Specific comments:

-Could the authors define "topobathymetric" LiDAR? I know that some manufacturers advertise their systems as "topobathymetric", but it was my understanding (and I may be wrong) that LiDAR systems operating within near-infrared wavelengths are "topographic", that those operating within the green wavelengths are "bathymetric", and that the multi-spectral systems that combine both near-infrared and green wavelengths are "topobathymetric". According to these definitions, the LiDAR used in this study would be a bathymetric LiDAR, although used both above and under the water surface. It would help if the authors specify/clarify why their system is considered topobathymetric while it is only surveying in the green wavelengths. This could be expanded on p. 2, after lines 22-24.

-The technical word for a digital model that considers both elevation and depth would be Digital Terrain Model. I suggest that you use DTM rather than DEM. I also recently read that some authors prefer using Coastal Terrain Model (CTM) for seamless models. It is sometimes ambiguous when the authors write "elevation", especially when underwater and talking about "depth". I suggest revisiting the use of these terms and using a neutral term such as "altitude" when needed, for instance on page 9, line 4.

-Page 3, lines 18-20: This is not necessarily true as some systems adjust the scanning angle to allow for a constant laser beam footprint. This was done for instance in Costa et al. (2009, vol. 113, Remote Sensing of Environment). It could be useful to specify.

-Page 5, lines 11-13 should come after line 4 (within that bullet point) as it describes study site 1.

-Are study sites 2 and 3 really study sites, or validation sites? I feel like the paper focusses on study site 1 rather than the study sites 2 and 3, which are only used to measure accuracy and precision. I would recommend changing the term "study site" for "validation site".

-Study site 3 is a steel frame located in a nearby river just below the water surface and is only used for precision assessment underwater. I wonder how representative of

the surveyed environment this study area really is. It seems to me that the river will present different environmental characteristics (e.g. turbidity, wave action, currents) than the main submerged area surveyed (i.e. flood channel). Also, would the precision measurement be different if the frame had been located deeper since light penetration is influenced by depth? These elements should be discussed at some point.

-Page 5, lines 23-26: What about the frame? It seems that if the frame is just below the water, a GPS could still be used to measure ground control points, as long as the pole on which the receiver is mounted is long enough to go on the frame without submerging the receiver.

-There was more than one month between the survey that assessed the accuracy and precision and the actual survey. Are there any potential implications? Was the surveying system used for different purposes in between these surveys? Were the environmental conditions similar enough?

-Page 6, lines 3-4: Could you provide a reference for this? Many surveyors would consider waves of half a meter high as not ideal conditions for surveying. Waves are known to influence the angle of penetration into the water, which then influences the way light refracts in the water column.

-Page 6, lines14-15: Table 1 summarizes the specifications of the LiDAR system, but are these corresponding to the survey characteristics? In other words, please specify if you actually surveyed at 400 m high that corresponded in a 400 m swath width, etc.

-Page 6, section 3.2: Please describe how these steps are different from the processing steps of any other surveys?

-Page 8, lines 14-15: The authors wrote that "based on visual inspection of the outcomes, it was impossible to reach a setting which would be optimal for all the different environments." However, at lines 22-23, they say that "based on the visual inspection of the filtering sensitivity analysis, the chosen settings for the automatic filtering were:…"

If it was first considered impossible, how were the settings actually chosen? What are the implications of selecting settings that are not optimal for all environments? Does it only influence the level of manual editing needed after?

-Page 9, lines 16-17: It is indicated here that the maximum wave heights were 20-30 cm while it was mentioned earlier that they were 50 cm. Please clarify.

-Page 9, lines 17-18: I think it would be preferable to provide a reference that indicates that it was actually acceptable to assume a flat water surface, or elaborate on why it is acceptable.

-What are the implications of selecting the higher value of the point cloud for the water surface and the lowest for the seabed? As mentioned previously, this goes against what would be considered by hydrographers as the basis for bathymetric data processing that serves their purpose. I am sure that the proposed method is suitable for other applications, but these would need to be discussed.

-It seems to me that the refraction correction could/should be done before the automatic and manual filtering. In multibeam data processing for instance, the sound velocity correction is performed before the cleaning of the soundings. Since the refraction correction is influencing the real 3D positioning of the LiDAR points, it would seem appropriate to start with the correction and then filter the points; some points that were outliers before the correction may simply have been more refracted and come closer to the rest of the points once corrected. Please justify the course of action and/or provide the appropriate references that justify your choice.

-Page 12, lines 18-21: Do you have any references to support the use of these planes as measures of accuracy and precision?

-Page 14, section 4.2: I am quite surprised that the precision was lower on land than underwater. Could you elaborate on that in the discussion? Also, I found unclear how the precision was actually measured for the frame, as the text focusses on the cement

block.

-Section 5.1: Considering the importance of the dead zone and as demonstrated in Figure 13, wouldn't have it been more effective to either survey at high tides, or to repeat the survey both at low and high tides in order to cover in one of the two passes what could not be covered by the other pass? I understand that my concerns are based on results (i.e. dead zone), but I think that you could develop in the discussion the implications of the dead zone, and possibly make recommendations for future surveys attempting to perform similar seamless coverage (c.f. the "take-home messages" that I mentioned in my general comment). Also in this section, I think that it would be interesting to discuss the implications of missing these "ponds". Depending on the potential applications of this dataset, the lack of information on these ponds can be meaningful. For instance, these ponds are likely important areas in the broader coastal ecosystem as they likely provide shelter or food to many species during low tides.

-Section 5.2: Most of the survey is actually above the water (topography rather than bathymetry). I believe that this may have an impact on the overall quality measurement of the dataset. Usually the accuracy is better when surveying land than the seafloor, and the accuracy is better in shallower waters than deeper waters. It would be interesting to discuss how the overall quality measurement may have been impacted by the amount of surveyed area that was above and below the water.

-Page 16, line 22: This is the first mention of the 4.1 cm. I suggest that the authors mention it before and following this sentence refer to the table from which it was taken from.

-Pages 16-17, lines 29-9: Many more studies compared the pros and cons of bathymetric LiDAR and multibeam echosounder data, including their accuracies. The conclusions are often that there are inconsistencies between the depth values collected from each system, and that multibeam echosounders are usually more accurate. However, I do not understand why the authors discuss these differences here considering

the study has nothing to do with multibeam sonars and that the argument that seems to be made is that multibeam is more accurate... That discussion does not support the results, on the contrary.

-Page 18, lines 9-15: I remember reading that slope does indeed reduce accuracy of LiDAR-retrieved bathymetry. The authors however need to support this affirmation with a reference. Also, is it possible to actually measure slope across the DTM and spatially compare slope and accuracy measurements? This would also relate the paper more to geomorphometry.

-Page 18, lines 21-23: How was that assessed? This affirmation (i.e. that LiDAR measurements are less precise in the channels) is not explained. How was it demonstrated? This also seems inconsistent with the fact that the frame (underwater) had a higher precision than the block (on land). How is that explained?

-Page 19, lines 3-5: "While bridging between spatial scales...". I do not understand this. It is the first mention of scale in the paper, and I do not see how this is a multiple scale analysis. Also, high accuracy does not necessarily result in high level of detail. A broad-scale dataset with low level of detail can have a very high accuracy, while a fine-scale dataset with a lot of details can have a very low accuracy.

-Pages 19-20, lines 30-2: I think that the differences and spatial variations of data quality throughout the study area are largely influenced by the differences in the environments (i.e. on land, on the tidal flat, in the channels and in the river) rather than simply the overlap between the swaths (although I agree that it is likely a factor).

-Page 20, line 10: Landscape needs to be defined as it can mean very different things for different persons, depending if they are remote sensing experts, landscape ecologists or urban planners.

-It could be relevant to calculate and add the kd factor of the area, measured with the attenuation coefficient and the maximum water depth. That could help explain the 3 m

maximum water depth that was achieved.

-What are the implications of the dead zone for particular applications? What are the implications of surveying only a very small proportion of the main channel, at a maximum depth of 3-4 m, when its average depth is of 15 m? This comes back to setting the paper into context. To which kind of applications would this particular dataset be useful?

Technical corrections:

-Page 5, line 19: "study site 2 and 3 was covered" should be "study sites 2 and 3 were covered".

-Page 8, in "automatic filtering": missing a reference for Fig. 4c.

-Page 19, line 2 and elsewhere: I suggest using "fine" and "broad" or "coarse" to characterise scale, as opposed to "small" and "large". Small and large scales are ambiguous as they have different meaning for different fields and professions.

-Page 21, line 29-30: This reference should be Klemas (2013) and not Klemas (2012).

-Add a space after the ";" in references listing within the text.

Potentially useful literature:

-Fernandez-Diaz, J.C., Glennie, C.L., Carter, W.E., Shrestha, R.L., Sartori, M.P., Singhania, A., Legleiter, C.J., Overstreet, B.T., 2013. Early results of simultaneous terrain and shallow water bathymetry mapping using a single wavelength airborne lidar sensor. IEEE J. Sel. Top. Appl. Earth Observations Remote Sens. 7 (2), 623–635.

- Allouis, T., Bailly, J.S., Pastol, Y., Le Roux, C., 2010. Comparison of lidar waveform processing methods for very shallow water bathymetry using Raman, nearinfrared and green signals. Earth Surf. Proc. Land. 35 (6), 640–650.

-Klemas, V., 2011. Remote sensing for studying coastal ecosystems: an overview.
[Figure]

Journal of Coastal Research, 27, 2-17.

---

## Referee Comment (RC2) · Anonymous Referee #2 · 17 Mar 2016

The authors presented a method of processing topobathymetric LiDAR data, and the accuracy of the production of DEM data is compared to others. Generally, it is an interesting and meaningful research. But there are some major aspects that should be process. Such as, the development of the method mentions little. I do not know the existing methods, let alone the shortcomings and advantages. At some extents, it looks like a technical report. I strongly suggests the authors improve the sections of "Introduction" and "Discussion". Specifically, Table 1 is not necessary; the numbers of the figures is a little high, and some has low quality. I suggest the authors can reduce some figures that are not necessary.

The answers of the questions for guiding the reviewers are the following:

1) Does the paper address relevant scientific questions within the scope of HESS?

[Figure]

Seldom. The relevant literature is scarce, and the analysis should be strengthen.

2) Does the paper present novel concepts, ideas, tools, or data? Few. I can not see much in the manuscript.

3) Are substantial conclusions reached? A little. The authors give specific procedures about the method, but the novel and development should be illustrated.

4) Are the scientific methods and assumptions valid and clearly outlined? Yes, the methods are illustrated clearly.

5) Are the results sufficient to support the interpretations and conclusions? Almost. The accuracy of the results is OK.

6) Is the description of experiments and calculations sufficiently complete and precise to allow their reproduction by fellow scientists (traceability of results)? I think so. The authors made abundant studies, and I infer that it can be reproduced.

7) Do the authors give proper credit to related work and clearly indicate their own new/original contribution? Probably not. The relevant literature is not limited, and the comparisons are not enough.

8) Does the title clearly reflect the contents of the paper? Yes.

9) Does the abstract provide a concise and complete summary? Yes.

10) Is the overall presentation well structured and clear? Yes.

11) Is the language fluent and precise? Yes.

12) Are mathematical formulae, symbols, abbreviations, and units correctly defined and used? Yes.

13) Should any parts of the paper (text, formulae, figures, tables) be clarified, reduced, combined, or eliminated? I suggest the figures can be reduced, whose quality improved, and Table 1 eliminated.

14) Are the number and quality of references appropriate? I strongly suggest adding the literatures, and analyze them specifically ad give deep comparison with this paper.

15) Is the amount and quality of supplementary material appropriate? There are not supplementary materials in this manuscript.
* * *

---

## Referee Comment (RC3) · Anonymous Referee #3 · 28 Mar 2016

The authors present a method for processing LiDAR data of a costal area, integrating land topography and shallow-water bathymetry. After the extensive and complete review of reviewer 1, I feel that much of what I say will sound repetitive. The paper feels much like a technical report. The authors claim to present a novel method but im my opinion fail to make clear in which ways their method is new, since pretty much all of the methodology is derived from other papers. Maybe they intent to say that the whole workflow is the novelty? As it is, the paper is more of an 'application paper' or a 'case study' than a scientific paper. The list of references is quite short, and much of the theoretical basis if from a textbook rather than papers.

**Does the paper address relevant scientific questions within the scope of HESS?**
The issue of processing topobathy LiDAR is of interest, but the paper format is not really suited to HESS.

[Figure]

**Does the paper present novel concepts, ideas, tools, or data?** Like I said, much of the methods are from others, and the authors don't make clear what is new, what is their contribution.

**Are substantial conclusions reached?** Not really. The conclusions are consistent with what is presented, but the authors should explore more the data they have and both the limitations of the method and how their results can be considered better than with other methods.

**Are the scientific methods and assumptions valid and clearly outlined?** Yes

**Are the results sufficient to support the interpretations and conclusions?** The conclusions are basically a report of the accuracy/precision of the final product, not from a deeper discussion of the data/experiments/results.

**Is the description of experiments and calculations sufficiently complete and precise to allow their reproduction by fellow scientists (traceability of results)?** Well, any 'manual interpretation' (like filtering) is hard to reproduce.

**Do the authors give proper credit to related work and clearly indicate their own new/original contribution?** Not really. There should be more references from scientific papers and less from textbooks and 'grey literature' (meetings abstracts, thesis...).

**Does the title clearly reflect the contents of the paper?** It would be more clear if the authors used something like 'case study'.

**Does the abstract provide a concise and complete summary?** It doesn't state the objectives or methods, just a brief introduction and the results.

**Is the overall presentation well structured and clear?** I don't think so. The authors should re-structure the paper so it becomes clearer what is their contribution to the processing workflow.

**Is the language fluent and precise?** Yes, it's ok.

[Figure]

**Are mathematical formulae, symbols, abbreviations, and units correctly defined and used?** Yes. I'm not sure if they are all necessary (the formulae for refraction of laser beams, form instance: it's mentioned but not really used, since a proprietary software/algorithm is used for that step of the processing).

**Should any parts of the paper (text, formulae, figures, tables) be clarified, reduced, combined, or eliminated?** Some formulae could be eliminated, I guess.

**Are the number and quality of references appropriate?** Not really. See above.

**Is the amount and quality of supplementary material appropriate?** This paper doesn't have supplementary material.

---

## Short Comment (SC1) · 31 Mar 2016

*Author's response to Referee #1*

We would like to thank you for your comments and very constructive suggestions. Your feedback on the manuscript is very valuable, and it will help us to improve the manuscript. We have been discussing thoroughly and extensively your suggestions and criticism, and therefore it has taken some time to respond.

The original project work included a technical part (as presented in the present manuscript) as well as a geomorphological part with a morphometric analysis of the test site. We decided to focus on the technical part in order to provide all details for the community, which is still emerging in the field of airborne green laser scanning and imaging. However, based on your comments and suggestions we will include the morphometric analysis in order to demonstrate the application for mapping morphological units in high energy intertidal environments, and specifically in relation to the vast intertidal flats in the Wadden Sea, which are otherwise impossible to map with full coverage in high detail.

The general comments and the evaluation points deal with some overall issues. Since these issues are first introduced in the general comments and then elaborated on in one or more of the evaluation points, we have decided to address each issue separately rather than addressing every point. We find that the issues addressed in the general comments and in the evaluation points can be summed up under the following headings:

1) Novelty

We acknowledge that it can be difficult to grasp the novelty of our proposed processing method. After all it is not the first time that a seamless DEM across the land-water transition zone has been derived from green LiDAR, and the main processing steps (filtering, water surface detection, refraction…) are unavoidable when processing such datasets. However, we do see our manuscript as being novel in these following ways (which we will work on clarifying):

- The water surface detection method is new, and it relies on simple concepts, which we have not seen in other studies. We will make this clear by referring to existing water surface detection methods in the manuscript (this links to issue #5). However, it is our experience that detailed descriptions of water surface detection methods, which focus on deriving the water surface from only a green LiDAR dataset (no NIR LiDAR data), are indeed very rare to come across.

- The entire data processing method has never (from what we know) been openly described to such a high level of detail, which makes the processing repeatable. Of course the commercial LiDAR companies have their workflows but there is always a tendency to keep the minor steps in their workflow hidden. Therefore we argue that our manuscript in the very detailed description of the workflow provides new knowledge to a broader audience.

- As already mentioned, we have decided to include a morphological classification part in the manuscript, based on the processed DEM. By this, we will add context to the manuscript, which will be novel in the sense that we make our own composition of classification tools and criteria (this is elaborated under issue #2).

**2) Morphological quantitative measurements**

We are aware that the manuscript is very technical in its present state, with the focus on the data processing and quality assessment. Even though we did not show an application of the processed DEM, we have, as already mentioned, worked on a morphological classification analysis tailored to the sandy intertidal flats of the Wadden Sea. We acknowledge that the present manuscript will be improved by adding a scientific (morphometric and morphological) analysis; and, moreover, making this addition will put the data collection and processing into an application context. Therefore we will add the morphological classification to the present manuscript.

**3) Context**

The technical part of data processing will be put in an application context by adding the morphological classification analysis (issue #2). With this addition, we will clarify the reason for data collection and processing.

**4) Extended discussion**

We acknowledge that we can improve the discussion by elaborating more on the scientific implications of our data collection, processing and results as well as on the state-of-the-art within water surface detection. We will modify the discussion so it includes:

- The implications of our LiDAR data collection and processing in the context of a morphological classification application.
- Implications of the dead zone and recommendations for future surveys.
- A comparison between our water surface detection methods and existing methods.

**5) More references to relevant literature**

We will discuss our water surface detection method further against existing methods and in doing so we will refer to more relevant literature within this subject. Besides, we will generally put further effort in supporting our claims with references to peer-reviewed papers.

**6) "Lessons learned"**

We think that it is a good recommendation and that it will improve the paper if the lessons learned (recommendations for future surveys) are expressed in a clear way. We will implement this in the discussion.

We find these six points to be the major concerns outlined in the review. The modifications involved in dealing with the issues require some major amendments to the manuscript, but thereafter it will be improved by being more scientific, novel and with a clear context of application.

We are currently working on addressing all the specific comments and reorganizing the manuscript, while we await the reply from the editors.

Once again thank you for your time reviewing the manuscript. We would look forward to submitting a revised version addressing and incorporating your comments and suggestions.

---

## Short Comment (SC2) · 31 Mar 2016

Thank you for taking the time to review our manuscript and for your comments and suggestions. Overall these are enclosed in the review by Referee #1; hence, please refer to our extended reply to Referee #1. We are currently working on addressing all the specific comments and reorganizing the manuscript, while we await the reply from the editors; and we would look forward to submitting a revised version addressing and incorporating your comments and suggestions.

---

## Author Comment (AC1) · 3 May 2016

Final author comments:

We have in the meantime, as mentioned in the previous author's response to the referees, addressed the issues and specific comments raised by the referees, and reorganized the manuscript. We have included the morphometric analysis in order to demonstrate the application for mapping morphological units in high energy intertidal environments, and specifically in relation to the vast intertidal flats in the Wadden Sea, which are otherwise impossible to map with full coverage in high detail.

We have analysed the DEM using a modification of the tool Benthic Terrain Modeler (BTM). Initially, stage (the elevation in relation to tidal range) was used to divide the area of investigation into the different tidal zones, i.e. subtidal, intertidal and supratidal.

Subsequently, morphometric units were identified and characterised by a combination of statistical neighbourhood analysis with varying window sizes (using the Bathymetric Positioning Index (BPI) from the BTM, moving average and standard deviation), slope parameters and area/perimeter ratios. Finally, these morphometric units were classified into six different types of landforms based on their stage and morphometric characteristics, i.e. either subtidal channel, intertidal flat, intertidal creek, linear bar, swash bar or beach dune.

We hereby demonstrate the potential of using airborne topobathymetric LiDAR for seamless mapping of land-water transition zones in challenging coastal environments with high water column turbidity and continuously varying water levels due to tides. Furthermore, we demonstrate the potential of morphometric analysis on high-resolution topobathymetric LiDAR data for automatic identification, characterisation and classification of different landforms present in coastal land-water transition zones.

Finally, this paper addresses one of the main themes outlined in the review paper of this issue "Characterising the ocean frontier: A review of marine geomorphometry" under the section "The future of marine geomorphometry", namely the filling of the gap between land and water. Hence, we remain arguing that the paper is of relevance for the community, and we believe that it is valuable to present the complete processing line from raw LiDAR data processing to automatic landform classification based on a geomorphometric analysis.

With kind regards on behalf of all authors,

Verner B. Ernstsen

---

## Author Response (AR1)

**Authors' response to the reviews**

Once again we would like to thank all of the referees for their comments and suggestions. We have been studying and discussing thoroughly your sharp criticisms and comments which have helped us to identify sections in the manuscript that require supplementing, rewriting or improving. The general referee comments and the evaluation points can be categorised into six major issues. Since these issues are first introduced in the general comments of Referee #1 and then elaborated in one or more of the evaluation points, we suggest to address each issue separately rather than replying to every point. We find that the issues addressed in the general comments and in the evaluation points can be summed up under the following headings together with the changes we have made to the manuscript accordingly:

**1. Novelty**

One of the major criticisms put forth by the referees was the difficulty of grasping the novelty of our proposed processing method. After all it is not the first time that a seamless DEM across the land-water transition zone has been derived from green LiDAR, and the main processing steps (filtering, water surface detection, refraction…) are unavoidable when processing such datasets. However, we do see our presented work as novel in the ways mentioned below, and we have clarified this in the manuscript:

- We have presented a simple procedure for water surface detection/modelling in the coastal zone using only green LiDAR data. To the authors' knowledge, such processing procedure has not been published before. We have referred to existing published articles that deal with LiDAR data water surface detection/modelling; however, most often NIR LiDAR data are used for the water surface detection, and the few studies dealing with green LiDAR data water surface detection do not go into detail on how to perform the actual modelling of the water surface in a coastal environment.
- The entire method for processing raw green LiDAR data into a DEM has never (to our knowledge) been openly described to such a high level of detail, which makes the processing useful, user friendly, and repeatable. The commercial LiDAR companies have their processing workflows but there is a tendency to keep some steps in their workflows hidden. Therefore we argue that our manuscript in the very detailed description of the workflow provides new knowledge to a broader audience. We have made this clear in the manuscript.
- We have developed a morphological classification based on the processed DEM, thereby adding scientific context to the manuscript (see point 3). We have made our own composition of tools and criteria for the morphological classification, and with this addition we have built a processing procedure that extends all the way from the raw data to classes of morphological features in a coastal environment.

**2. Context**

A second major issue, addressed by the referees, was the lack of context to the manuscript, and they were absolutely right. Our first submission focused on the technical part of data processing; however, the original project work included also a geomorphological part with a morphometric analysis of the test site. We decided to focus on the technical part in order to provide all details for the community, which is still emerging in the field of airborne green laser scanning and imaging. However, based on the referee comments and suggestions we have decided to include the morphometric analysis in order to demonstrate the application for mapping morphological units in high energy intertidal environments, and specifically in relation to the vast intertidal flats in the Wadden Sea, which are otherwise impossible to map with full coverage in high detail. This addition will further clarify the reason for data collection and processing, which was also an issue addressed by the referees.

**3. Morphological quantitative measurements**

It was criticized that the originally submitted manuscript was lacking morphological quantitative measurements. We have overcome this issue by adding the morphometric analysis and morphological classification as mentioned in the previous point.

**4. Extended discussion**

We have consolidated the discussion by relating our developed processing procedure with the state-of-the-art methods of water surface detection and by elaborating more on the scientific implications of our data collection, processing and results. Specifically, we have modified the discussion so it includes:

- A comparison between our water surface detection methods and existing methods.
- The implications of the dead zone.
- The implications of using LiDAR data collected at different days and in different environments for data processing and quality assessment, respectively.
- Our LiDAR data processing method in the context of a morphological classification and the method's transferability to other applications.
- Evaluation and potential of using topobathymetric LiDAR data for mapping morphological features in a highly dynamic tidal environment.

**5. More references to relevant literature**

The quality and types of references were criticized by the referees in our first submitted manuscript. We have addressed this by including discussion of our method for water surface detection against up to date, peer reviewed published literature on the subject.

**6. "Lessons learned"**

We addressed this issue in quite few locations in the manuscript when we described the water surface modelling method in detail giving the current achievement and the future work required for enhancing the modelling accuracy via incorporating the wave and slope models in the workflow. We also discussed the choice of thresholds in the morphological classification and showed the importance of choosing the right thresholds for producing the actual morphological features, and concluded that an objective method is required in future work to estimate these thresholds, which renders the method applicable in all environments. We have also demonstrated the ability of green LiDAR to map seamlessly the land-water transition zone with such high accuracy and precision to make it a practical and excellent choice for conducting such work in the coast zone.

We found these six issues to be the major concerns outlined by the referees. The modifications involved in dealing with the issues have required some major amendments, but we are confident that the manuscript has improved.

Thank you for your time reviewing the manuscript and for your suggestions for improving it. We look forward to receive your response to these improvements.

On behalf of all authors,

Faithfully,

Verner B. Ernstsen
Associate Professor

[revised manuscript text omitted]
 is to investigate the potential of topobathymetric LiDAR data to accurately model the real world terrain and surface in land-water transition zones. The aim is achieved by meeting the following objectives:

The aim of this study was to investigate the potential of improving the processing procedure of green LiDAR data for generating DEMs in tidal coastal environments characterised by land-water transition zones, and of improving the classification of morphological units in such environments. More specifically, the objectives were:

1. To develop a robust, repeatable and user friendly processing procedure of raw green LiDAR data To develop a processing procedure for generating high resolution DEMs the generation of a digital elevation model DEM in land-water transition zones.

2. Quantify To quantify the accuracy and precision of the green LiDAR data based on object detection.

3. Evaluate the potential of topobathymetric LiDAR to resolve landforms in land-water transition zones.

3. To automatically classify morphological units based on morphometric analyses of the generated DEM.

The investigations were based on studies undertaken in a section of the Knudedyb tidal inlet system in the Danish Wadden Sea.

**2 Study area**

The Knudedyb tidal inlet system is located between the barrier islands of Fanø and Mandø in the Danish Wadden Sea (Fig. 2A). The tidal inlet system is a natural environment without larger influence from human activity. The tides in the area are semi-diurnal, with a mean tidal range of 1.6 m, and the tidal prism is in the order of $175 \cdot 10^6$ m$^3$ (Pedersen and Bartholdy, 2006). The main channel is approximately 1 km wide and with an average water depth of approx. 15 m (Lefebvre et al., 2013).

The study site is an elongated 3.2 km$^2$ (0.85 × 4 km) section of the Knudedyb tidal inlet system (Fig. 2B). The section is located perpendicular to the main channel and stretches across both topography and bathymetry. The study site extends towards north into an area on Fanø with dispersed cottages (Fig. 2C). The most prominent morphological features within the study site include beach dunes (Fig. 2D), small mounds (Fig. E), swash bars (Fig. 2F-G) and linear bars (Fig. 2H).The quality of the LiDAR data were validated at two sites along Ribe Vesterå River (Fig. 2I-J):

- Validation site 1 is a cement block with a size of 2.50×1.25×0.80 m located on land next to the mouth of Ribe Vesterå River (Fig. 2I). The block was used for assessing the accuracy and precision of the LiDAR data.

- Validation site 2 is a steel frame with a size of 0.92×0.92×0.30 m located in the river with the surface just below the water surface (Fig. 2J). The frame was used for precision assessment., and for testing the feature detection capability of the LiDAR system. According to the hydrographic survey standards presented by the International Hydrographic Organization (IHO, 2008), cubic features of at least 1 m$^2$ should be detectable in Special Order areas, which are areas with very shallow water as in the study site .

**3 Methods**

**3.1 Surveys and instruments**

LiDAR data and ortophotos were collected by Airborne Hydro Mapping GmbH (AHM) during two surveys on 19 April 2014 and 30 May 2014.

On 19 April 2014,  validation sites 1 and 2 were covered for accuracy and precision assessment of the LiDAR data by object detection of the block and the frame (for location see Fig. 2). The block was covered by 7 swaths retaining 227 LiDAR points from the block surface. The frame was covered by 4 swaths retaining 46 LiDAR points from the surface of the frame. Ground control points (GCPs) were measured for the four corners of the block with accuracy better than 2 cm using a Trimble R8 RTK GPS. Measurements were repeated three times and averaged to minimize errors caused by measurement uncertainties. GCPs were also collected for the frame; however, during the LiDAR survey the frame experienced an unforeseen intervention by local fishermen using the frame as fishing platform. Therefore, the frame is only used to assess the deviation between the LiDAR points (the precision), and not to assess the deviation between the LiDAR points and GCP's (the accuracy).

On 30 May 2014, the study site  was covered by 11 swaths , which were used for generating the DEM. Low tide was -1 m DVR90, measured at Gråddyb Barre, approx. 20 km NW of the study site.

The weather conditions were similar during the two surveys, with sunny periods, average wind velocities of 7-8 m/s (DMI, 2014) and approx. 0.5 m wave heights coming from NW, measured west of Fanø (DCA, 2014). The wave heights in the less exposed Knudedyb tidal inlet was observed in the LiDAR data to 0.2-0.3 m. Overall, both days constituted good conditions for topobathymetric LiDAR surveys.

In both surveys, LiDAR data were collected with a RIEGL VQ-820-G topobathymetric airborne laser scanner. The scanner is characterized by emitting green laser pulses with 532 nm wavelength

and 1 ns pulse width. It has a very high laser pulse repetition rate of up to 520,000 Hz and, and a beam divergence of 1 mrad creates a narrow laser beam footprint of 40 cm diameter at a flying altitude of 400 m (RIEGL, 2014).(RIEGL, 2014), which was the actual flying altitude during the surveys. The high repetition rate and narrow footprint makes it well suited to capture smallfine-scale landforms (Doneus et al., 2013;Mandlburger et al., 2011;RIEGL, 2014)(Doneus et al., 2013; Mandlburger et al., 2011; RIEGL, 2014). An arc shaped scan pattern maintainsresults in a swath width of approx. 400 m (at 400 m flying altitude), while maintaining an almost uniform scan angle of constant 20° (±1°), which is influenced by the roll, pitch and yaw of the airplane. This means that the°) incidence angle of the laser beam is almost constant at when it penetrates the water surface (Niemeyer and Soergel, 2013)(Niemeyer and Soergel, 2013). General specifications of the laser scanner are summarized in Table 1 (RIEGL, 2014;Steinbacher et al., 2012). The typical water depth penetration of the laser scanner is 1 Secchi disc depth.

For each returned signal, the collected LiDAR data contained information of x, y and z, as well as a GPS time stamp and values of the amplitude, reflectance, return number, attribute and laser beam deviation (RIEGL, 2012). Primarily the positions and time stamps of the LiDAR points were used in the data processing. The reflectance, which represents the range normalized amplitude of the received signal, was used to a lesser extent in the filtering process.(RIEGL, 2012).

**3.2   FromProcessing raw topobathymetric LiDAR data tointo a gridded DEM**

A list ofThe essential processing steps was necessary to produce a DEM from raw, which are standard procedure when processing topobathymetric LiDAR data., were followed to produce a DEM in the study area. These steps included:

1. Determination of flight trajectory.

2. Integration of sensor data (laser scanner data, motion sensor data, positioning/trajectory data).

3. Raw point cloud processing.

[revised manuscript text omitted]
). ~~For future investigations it will be an improvement if all the water surfaces are modelled. This could be achieved by implementing NIR LiDAR measurements in the LiDAR survey, since it is reflected by any water surface. It may also be achieved with green LiDAR as the only data source by detecting the returned signals reflecting off the water surface in the dead zone. Potentially, this could be achieved by analysing the waveforms and choosing the first local peak in the returned signal as a valid detected point. Thereby, both the sea bed and the water surface would have a seamless transition between land and water.~~

**5.3 Evaluation of the topobathymetric LiDAR data quality**

The vertical accuracy of conventional topographic LiDAR has previously been determined to ±10-15 cm (Hladik and Alber, 2012; Jensen, 2009; Klemas, 2013; Mallet and Bretar, 2009). Only few previous studies have focused on the accuracy of shallow water topobathymetric LiDAR data (Mandlburger et al., 2015; Nayegandhi et al., 2009; Steinbacher et al., 2012). Nayegandhi et al. (2009) determined the vertical $E_{RMS}$ of LiDAR data in 0-2.5 m water depth to ±10-14 cm, which is above the ±4.1 cm $E_{RMS}$ found in this study (Table 1). Steinbacher et al. (2012) compared topobathymetric LiDAR data from a RIEGL VQ-820-G laser scanner with 70 ground-surveyed river cross sections, serving as reference, and found that the system's error range was ±5-10 cm, which is comparable to the ±8.1 cm accuracy found in this study. Mandlburger et al. (2015) compared ground-surveyed points from a river bed with the median of the four nearest 3D-neighbors in the LiDAR point cloud, and they found a standard deviation of 4.0 cm, which is almost

equal to the ±4.1 cm standard deviation found in this study (Table 1). In comparison with these previous findings of LiDAR accuracy, the assessment of the vertical accuracy in this study indicates a good quality of the LiDAR data.

Mapping the full coverage of tidal environments, such as the Wadden Sea, require a combination of topobathymetric LiDAR to capture topography and shallow bathymetry and MBES to capture the deeper bathymetry. The two technologies make it possible to produce seamless coverage of entire tidal basins; however, merging the two products raises the question whether the quality of the data from the two different sources is comparable. Comparing the LiDAR accuracy with previous findings of accuracy derived from MBES systems indicates similar or slightly better accuracy from the MBES systems (Dix et al., 2012; Ernstsen et al., 2006)~~. Dix et al. (2012) determined the vertical accuracy of a multibeam sonar by testing the system on different objects and in different environments, and found the vertical $E_{RMS}$ to be ±4 cm. Furthermore, they tested a LiDAR system on the same objects and found a similar vertical $E_{RMS}$ of ±4 cm. The vertical $E_{RMS}$ of ±4.1 cm found in this study is very close to both the multibeam accuracy and LiDAR accuracy determined by Dix et al. (2012). Another study by Ernstsen et al. (2006) determined the vertical precision of a multibeam sonar based on 7 measurements of a ship wreck from a single survey. They found the vertical precision to be ±2 cm, which is slightly better than the vertical precision of ±3.8 cm (frame) and ±7.6 cm (block) found in this study.~~

~~Determining vertical accuracy and precision are standard practice in studies involving spatial data (FGDC, 1998;Graham, 2012;Jensen, 2009). Accuracy and precision are in many cases provided as single values, such as ±8.1 cm for the vertical accuracy in this case, and thereafter they represent the accuracy/precision of the whole dataset. However, the values actually only apply to the specific locations, where the assessment is conducted. In reality, the accuracy and precision may vary spatially, which is also the case by the differing precision of ±3.8 cm at the steel frame and ±7.6 cm at the cement block in this study. Furthermore, spatial variations of the precision throughout the study area are revealed by looking at the vertical difference between overlapping LiDAR measurements (Fig. 14).~~

[revised manuscript text omitted]
 is evident by comparing number of swath overlaps (Fig. 9B) and the local point density (Fig. 9A) with the local vertical difference of the LiDAR points (Fig. 14).

This can be caused by variance/error in the GPS measurements and/or IMU errors (Huising and Gomes Pereira, 1998). The vertical bias between swaths is varying and it has been observed in the point cloud to be up to 5 cm., but it is varying throughout the study site. In most environments, a bias of 5 cm would be unnoticeable, but because of the large and very flat parts of the Knudedyb tidal inlet system, even a small bias becomes readily evident. The bias between overlapping swaths may explain the lower precision at the block compared to the frame, because the block was covered by 7 swaths as opposed to 4 swaths at the frame. It seems counterintuitive that more overlapping swaths, leading to higher point density, eventually result in *lower* precision of the measurements. In this case, the difference between precision and accuracy should be kept in mind, and that the same relationship between overlapping swaths and accuracy does not necessarily exist.

Sloping areas: LiDAR measurements on sloping areas are expected to have lower vertical accuracy than on flat ground, because the laser beam footprint may span across different elevations. The exact position of the detected point can vary within the footprint, and thus it may also vary in elevation. Furthermore, the slope affects the footprint by increasing its area size and changing the shape to more elliptical and less round. The influence of slope is not crucial in the Knudedyb tidal inlet system, since it is generally a very flat area, but it is still an uncertainty factor to keep in mind.

Uncertainty with increased water depth: The accuracy and precision are expected to be lower as the laser beam penetrates deeper into the water column. It is first of all due to widening of the laser beam footprint, which means that the elevation of a single LiDAR point is derived from the measurement on a larger area on the sea bed. Secondly, any uncertainty associated to a LiDAR measurement is magnified with increasing water depth, due to the refraction correction. These

factors, together with slopes, are causing the LiDAR measurements to be less precise in the main channel and in the flood channel.

Water depth: The accuracy and precision are expected to be lower as the laser beam penetrates deeper into the water column (Kunz et al., 1992). The laser beam footprint is diverging as it moves through the water column, resulting in a larger footprint on the seabed. The altitude of the detected point is thus derived from the measurement on a larger area on the seabed, which will decrease the vertical accuracy, as well as decrease the capability of detecting small objects. With this in mind, the lower precision at the frame compared to the block is opposite of what would be expected, since the frame is below water and the block is on land. In this case, other factors, such as overlapping swaths and/or scan angle deviations, have more influence on the precision than the water depth. Also, it should be remembered that the frame surface was close to the water surface, and the effect of the water depth on the precision would most likely be more evident if it was located in deeper water.

Additional factors, beside the ones mentioned above, may increase the uncertainty ofinfluence the quality of LiDAR datasets. This could for For instance be, a dense vegetation coveringcover of the ground or sea bed,seabed or breaking waves, which that makes it the laser detection of the seabed almost impossible for the laser to detect the sea bed. However, these factors do not have a great influence in the studied part of the Knudedyb tidal inlet system, and thus they are not further elaborated. Nevertheless, these factors must be taken into consideration for LiDAR surveys in different areas with lots of vegetation and slopes.

**5.35.5 ImpactEvaluation of the findingsmorphological classification**

The morphological classification presented in this study is based on the studied section of the Knudedyb tidal inlet system. The overall concept of using tidal range, slope and variations of the altitude at different spatial scales proves to be a reliable method for delineating the morphological features in this tidal environment. The concept, however, can be applied in other environments. The specific thresholds in the classification determined in this study may deviate in other areas. Morphological features of different sizes require steps of other spatial scales in the neighbourhood analyses to produce a successful classification. In the future the classification method will be improved by implementing an objective method for determining the scales, which can make it applicable in areas with different morphological characteristics. Such an objective scale

determination method is presented by Ismail et al. (2015), who determined the scales based on the variance of the DEM at progressively larger window sizes. In this way, the sizes of the morphological features are determining the scales for the classification.

**5.6   Using topobathymetric LiDAR data to map morphology in a highly dynamic tidal environment**

The study demonstrates the capability of green topobathymetric LiDAR to resolve smallfine-scale features, while covering a broad-scale tidal inlet system. Collecting topobathymetric LiDAR data with a high point density of 20 points/m$^2$ on average enables detailed seamless mapping of large-scale tidal inlet system. While bridging between spatial scales, tidal environments, and the LiDAR data has further proved to maintain a high accuracy, which means that shallow water zones can be mapped with a high level of detail.. The combined characteristics of mapping with high resolution and high accuracy in a traditionally challenging environment provide many potential applications to the society, such as mapping for purposes of spatial planning and management, safety of navigation, or nature conservation., or morphological classification, as demonstrated in this study. The developed LiDAR data processing method is tailored to a morphological analysis application. The best representation of the morphology is mapped by gridding the average value of the LiDAR points into a DEM with a $0.5 \times 0.5$ resolution. Other applications would require different gridding techniques. For instance hydrographers, who are generally interested in mapping for navigational safety, would use the shallowest point for gridding. However, the overall method for processing the point cloud can be used regardless of the application. Only the last and least challenging/time consuming step of gridding the point cloud into a DEM may vary depending on the application.

During a single LiDAR survey, the present state of the environment is captured with high resolution and high accuracy. However, the coastal zone is a highly dynamic environment influenced by complex hydrodynamic processes and feedback mechanisms. Therefore, a continuous monitoring of the coastal zone with high accuracy LiDAR systems will provide an insight to the temporal variation, whether caused by climate variation or inflected by human activities.

Applying topobathymetric LiDAR data for morphological analyses in tidal environments enables a holistic approach of seamlessly merging marine and terrestrial morphologies in a single dataset. In order to map the morphology of tidal environments in full coverage, however, a combination of topobathymetric LiDAR and MBES swath data is required. The comparable quality and resolution

of LiDAR and MBES data gives a potential to map large scale tidal environments, such as the Wadden Sea, in full coverage and with high resolution and high accuracy.

**6   Conclusions**

A new method was developed for processing raw topobathymetric LiDAR data into a digital elevation model with seamless coverage across the land-water transition zone. The point cloud processing is based on simple concepts, which are easily repeatable, and the processing steps are described in detail. The novel method Specifically a procedure was developed for water surface detection utilizing automatic water level determination from only green LiDAR data in a tidal environment. The method relies on basic principles, and in general the entire processing method is described with a high level of detail, which makes it easy to implement for future studies. Specifically, the The water surface is extrapolated, so that it also covers the "dead zone", which has been determined to be approx. 0-28 cm in the very shallow water. The method doesdetection method presented in this work did not model the spatially changing water levels, such as wavestake into account the variation in wave heights and inclined surfaces.surface slopes, which therefore constitutes a challenge to be addressed in future studies.

The vertical accuracy of the LiDAR data was determined by object detection of a cement block on land to ±8.1 cm with a 95% confidence level. The vertical precision was determined at the cement block to ±7.6 cm, and ±3.8 cm at a steel frame, placed just below the water surface. The difference between the two sites is an indication of spatial variations throughout the study area, largely influenced by biases between overlapping swaths. 
[revised manuscript text omitted]
 1: Specifications of the RIEGL VQ-820-G topobathymetric airborne laser scanner (RIEGL,
2  2014).

| | |
|---|---|
| Flight altitude | ~ 400 m above ground |
| Swath width | ~ 400 m |
| Scan pattern | Section of an ellipse – arc shape |
| Scan angle | 20° ±1° |
| Laser wavelength | 532 nm |
| Pulse width | 1 ns |
| Laser beam footprint (diameter) | 40 cm (at 400 m flight altitude) |
| Laser pulse repetition rate | Up to 520,000 Hz |
| Max. effective measurement rate | Up to 200,000 meas./sec. |
| Laser beam divergence | 1 mrad |
| Typical water depth penetration | 1 Secchi disc depth |

1 Table 2:

2  Table 1: Vertical accuracy and precision of the LiDAR point measurements, in terms of minimum

3  error ($E_{min}$), maximum error $(E_{max})$, standard deviation ($\sigma$), mean absolute error ($E_{MA}$), root mean

4  square error ($E_{RMS}$) and the 95% confidence level ($Cl_{95\%}$).

| Accuracy/ Precision | Object | Best-fit plane | # points n | $E_{min}$ (cm) | $E_{max}$ (cm) | $\sigma$ (cm) | $E_{MA}$ (cm) | $E_{RMS}$ (cm) | $Cl_{95\%}$ (cm) |
|---|---|---|---|---|---|---|---|---|---|
| Accuracy | Cement block | GCPs | 227 | 0.01 | 12.1 | 4.1 | 3.5 | ±4.1 | ±8.1 |
| Precision | Cement block | Point cloud | 227 | 0.04 | 12.9 | 3.9 | 2.8 | ±3.9 | ±7.6 |
| Precision | Steel frame | Point cloud | 46 | 0.02 | 5.5 | 2.0 | 1.6 | ±1.9 | ±3.8 |

1  Table 32: LiDAR point spacing and density for all the 11 individual swaths, which covered the
2  study area, and for the combined swaths.

| Swath number | 1 | 2 | 3 | 4 | 5 | 6 | 7 | 8 | 9 | 10 | 11 | All |
|---|---|---|---|---|---|---|---|---|---|---|---|---|
| Point spacing (m) | 0.30 | 0.30 | 0.36 | 0.31 | 0.36 | 0.32 | 0.37 | 0.29 | 0.35 | 0.36 | 0.28 | 0.20 |
| Point density (pt./m$^2$) | 10.8 | 10.8 | 7.8 | 10.2 | 7.5 | 9.6 | 7.2 | 11.7 | 8.0 | 7.8 | 12.7 | 19.6 |

[Figure]

Figure 1: Conceptual sketch of the laser beam propagation and return signals. The beam refracts upon entering the water body, and it diverges as it propagates through the water column. Return signals are produced both in the air, at the water surface, in the water column and at the seabed. The LiDAR instrument has limited capability in very shallow water (the "dead zone" in the figure) because the successive peaks from the water surface and the seabed are not individually separated in time and amplitude. Only the largest peak, which is from the seabed, is detected.

[Figure]

[Figure]

Figure 2: A) Overview of the study area location in the Danish Wadden Sea and the specific locations of the study site (B) and the two validation sites (I and J) three study sites (22 April 2015 satellite image, Landsat 8). B) StudyThe study site 1 in the Knudedyb tidal inlet system (30 May 2015 Orthophoto, AHM). C) Study site 2C) Cottages in the dunes on Fanø. D) Beach dunes on Fanø. E) Patch of *Spartina Townsendii* (Common Cord Grass). F-G) Swash bars. H) Linear bar. I) Validation site 1 with a cement block on land, used for accuracy and precision assessment (19 April 2015 orthophoto, AHM). D) StudyJ) Validation site 32 with a steel frame in Ribe Vesterå River, used for precision assessment (19 April 2015 orthophoto, AHM). E) Cottages in the dunes on Fanø. F) Beach dunes on Fanø. G) Patch of *Spartina Townsendii* (Common Cord Grass). H-I) Swash bars. J) Linear bar.

[Figure]

2 Figure 3: The 11 swaths covering study site 1, which were used for generating the DEM.

[Figure]

Figure 3: Workflow for processing the LiDAR point cloud. A) Point cloud from a single swath with points ranging from -100 m to 300 m elevation. B) Zoom-in on a cross section of the flood channel with altitudes exaggerated ×15 for visualization purpose. C-E) Method for filtering the point cloud. F-H) Method for detecting a water surface (blue) based on the extraction of a shallow surface (red) and a deep surface (orange). I) Correction for the effect of refraction on all the submerged points. J) Processed point cloud

[Figure]

2  Figure 4: Classification decision tree, showing how the geomorphometric classification was

3  conducted in the Benthic Terrain Model tool.

[Figure]

Figure 5: Classification decision tree of the morphological classification. All steps were performed in ArcGIS.

[Figure]

3 Figure 5:6: Vertical adjustment of the refracted LiDAR points from the flood channel transect (see

4 location in Fig. 2C).

[Figure]

Figure 6: Example of a cross section of the flood channel, with a clearly visible gap in the water points in the very shallow water. The vertical dead zone is determined to be approx. 28 cm (see text).

[Figure]

2 Figure 7: Vertical difference between the shallowest and the deepest LiDAR point within

3 0.5 m grid cells in the land-water transition zone.  The abrupt change is caused by the dead zone.

4 The vertical extent of the dead zone is determined to approx. 28 cm, derived by the maximum rate

5 of change of a polynomial fit through the points.

[Figure]

2 Figure 8:8: Vertical and horizontal distribution of the LiDAR points describing the block surface

3 and the block surface derived from Ground Control Points (GCPs). A) LiDAR points (grey dots)

4 compared to the GCP block surface (black line) for determining the vertical accuracy. The grey line

5 shows the LiDAR block surface as a best-linear-fit through the points. B) Block surface derived

6 from the four GCP corner points and the block surface derived by the perimeter of the LiDAR

7 points.

[Figure]

Figure 9: A) Point density (pts./m$^2$) throughout the study site. B) Number of swath overlaps in different sections of the study site.

[Figure]

2 Figure 10: Frequency distribution of the varying LiDAR point density throughout the study area.

[Figure]

Figure 9: Topobathymetric DEM across the northern part of the Knudedyb tidal inlet system with close-up views of different detail level on specific areas. A hill shade is draped upon the close-ups for improved visualization of morphological features. A) Northern section with beach dunes and cottages. B) Cottages. C) Mid-section with the flood channel. D) Closer view on an intertidal creek. E) Southern section with swash bars, linear bars and bathymetry of the main channel. F) Swash bar.

[Figure]

[Figure]

Figure 10: Two classifications of the investigated section in Knudedyb tidal inlet system, derived from a topobathymetric DEM. A) Geomorphometric classification. B) morphological classification. C) Zoom-in on the intertidal creek in the morphological classification. D) Zoom-in on the swash bars and linear bars close to the main channel in the  morphological classification. A

hillshade of the  DEM is draped over C and D.

[Figure]

Figure 11: Vegetated mounds on the intertidal flat.  are clearly  visible in the DEM and classified as small-scale crests in the geomorphometric BTM classification. To the right is an image of one of the patches.

[Figure]

2 Figure 13:12: Horizontal extent of the dead zone in the studied area at mean low water, mean water

3 level and mean high water.

[Figure]

Figure 13: Vertical difference between the highest and the lowest LiDAR point within 0.5  ×0.5 m grid cells.

---

## Referee Report (RR1)

The authors have done an excellent work in revising this manuscript based on the comments from the three reviewers. Most of my earlier concerns have been addressed by the authors in this revised version. The manuscript is now situated in context, with appropriate referencing, and the novelty is clearer. The discussion is better and the application (morphological characterization) makes it a more complete paper, suitable for publication in HESS. I only have a few comments listed below that I am sure the authors will be able to address.

1. I would suggest reducing the length of the title, for instance by removing "in the coastal zone" since the mention of "high-energy tidal environment" already suggests that it is in a coastal environment.

2. In the abstract (p. 1 line 17), I would replace the word "harsh" by either "difficult" or "challenging". I feel like these have a different meaning and environmental conditions are not necessarily harsh, but can be challenging for surveying. The authors use "challenging" to describe them in the main text, and the abstract should be consistent with the main text.

3. As noted in the previous round of reviews, I recommend not using digital elevation model (DEM) when considering both elevation and depth like in this study. Since the data include cottages and vegetation, I believe that the most appropriate term would be Digital Surface Model (DSM), as opposed to Digital Terrain Model (DTM) that represents a "bare-earth" model.

4. Also noted before: "landscape" has different meanings depending on the field of study (e.g. in landscape ecology or remote sensing). I recommend removing the two instances on page 17 and replacing them by "terrain".

5. Another one noted before: change "small-scale" and "large-scale" for "fine-scale" and "broad-scale" throughout the text and in figures (e.g. Fig. 4). The formers have different meanings in cartography and other fields than in this study. The latters are less ambiguous.

6. On page 8 (lines 3 to 14), this should be moved to the methods section below, likely between lines 19 and 20 of that same page. At line 9, I would change the first "surface" for a word like "top" to make it clearer, e.g. "…located in the river with its top just below the water surface."

7. I understand how the method that is described in this manuscript is more transparent, reproducible and user-friendly than current alternatives. However, I am unsure of the level of reproducibility for two reasons. First, while the authors use RiHYDRO, HydroFusion and LiDAR Survey Studio as examples for describing the lack of transparency in available software (p. 6), the proposed method still requires many software that may not be widely available for all users and are not necessarily more transparent (e.g. RiPROCESS, HydroVish, Fledermaus, MATLAB). Second, many steps involve manual processing (e.g. filtering, extracting the shallow surface) or subjective decisions (e.g. parameters for automatic filtering, classification trees, values of 4 for standard deviation to differentiate between features). I appreciate that the authors mention these limitations (e.g. p. 11 and 28), however I believe that care should be used when making claims like at page 22 ("it is open to the public"), especially since the software used may not be accessible to the public. I wouldn't go as far as suggesting the removal

of mentions of "reproducible", but I would be curious to see if this issue will also be of concern to other reviewers or the editor.

8. On page 13, line 11, what do the authors mean by "only taking the top 95-100% of water points into account"? If 100% of the points are considered, then they are all accounted for and can't possibly increase reliability. Please clarify.

9. On page 13, lines 16 to 31 are confusing. I am unsure what is the relationship between the 2 x 2 m water surface and the 0.5 x 0.5 m surface. I understand that the 2 x 2 m was built to remove outliers, but what is it used for then? This section requires rewording or clarifications.

10. On pages 16 and 17, please specify what the standard deviation represents (for the BPI). Is it the standard deviation of depth/elevation values within the window of analysis?

11. On page 17 (lines 3-4), I do not understand what the authors mean by "the altitude was exaggerated 10 times before the classification, to enable the BTM to detect the shapes of the landscapes". A vertical exaggeration in the visual representation of data would not change the altitude (depth or elevation) values and thus would not impact the results. If this is the case, then this sentence is unnecessary and can be removed. However, if the altitude values were actually altered and multiplied by a factor of 10, I am not sure if the analysis is still valid, although it would likely not change the relative values of pixels and still identify peaks and pits (but maybe reduce the amount of flat areas?). This needs to be clarified.

12. Page 17, line 5: "the best results" based on what? Visual interpretation? Was it a subjective decision? Please specify.

13. In Figure 5 and associated text, is everything >0.94 m really a beach dune? Weren't there cottages and other features? Would another term be more appropriate and all-encompassing of features that were above the water level?

14. On page 18, I believe that the window sizes are wrong. They would only make sense if a radius was used instead of a window of analysis. Window sizes need to be odd numbers and based on the pixel size (0.5 m), the window could not be of 100 m or 250 m wide. Please revise these measurements, indicate whether a window of analysis (square) was used or a radius (circle), and whether these numbers are the numbers of pixels of the window or the actual area covered by the window (in either cases it should be an odd number).

15. For your information, standard deviation (cf. classification trees) is used as a measure of rugosity in geomorphometry, so when the authors used it to distinguish between bars and larger features, they used a measure of broad-scale rugosity.

16. Page 18, lines 29-30: "4 were found to be a suitable ratio threshold". Based on what (e.g. visual interpretation, etc.)?

17. In Figure 10 (B-C-D), I would add a category in the legend to characterize the grey areas (that I assume are no data, i.e. the areas that did not correspond to any of the criteria in the decision/classification tree). This category could be named "Transition zones".

18. On page 25, lines 4 to 17 are repetitive to the methods section (p. 9). I would bring back the description of the environmental conditions at time of survey in the methods section, and keep the discussion on their implications at p. 25.

19. The use of English language could be improved before publication, although it is not a big concern at the moment as the text remains clear. For instance, p. 10 line 21 should read "Steps 5-8 represent" instead or "Step 5-8 represents", p. 12 line 14 "currently" should be removed since it was at time of survey, p. 18 line 29 "was" instead of "were", p. 29 line 18 "are" instead of "is", etc.. These are only a few examples of where corrections should be made. Also, in some parts of the text that describe the methods and results, the past tense should be used.

---

## Author Response (AR2)

**Authors' response to the Editor and the Referees**

Dear Editor, dear Referees,

We thank you for your time reviewing the manuscript and for your constructive comments and suggestions for improving the manuscript.

We have addressed all of the comments addressed in the reviews. Please find below a point-by-point response to the Review Reports as well as a marked-up manuscript version.

We await your decision on this revised version of the manuscript, and please contact us in case you require any additional information or clarification.

On behalf of all authors,

Sincerely,

Verner B. Ernstsen
Associate Professor
Department of Geosciences and Natural Resource Management
University of Copenhagen, Denmark

**Authors' response to Report #2 of Referee #1**

*Authors' response is italicized.*

R1: The authors have done an excellent work in revising this manuscript based on the comments from the three reviewers. Most of my earlier concerns have been addressed by the authors in this revised version. The manuscript is now situated in context, with appropriate referencing, and the novelty is clearer. The discussion is better and the application (morphological characterization) makes it a more complete paper, suitable for publication in HESS. I only have a few comments listed below that I am sure the authors will be able to address.

R1: 1. I would suggest reducing the length of the title, for instance by removing "in the coastal zone" since the mention of "high-energy tidal environment" already suggests that it is in a coastal environment.
> *The length of the title has been reduced as suggested.*

R1: 2. In the abstract (p. 1 line 17), I would replace the word "harsh" by either "difficult" or "challenging". I feel like these have a different meaning and environmental conditions are not necessarily harsh, but can be challenging for surveying. The authors use "challenging" to describe them in the main text, and the abstract should be consistent with the main text.
> *The adjectives describing the environmental conditions have been changed as suggested.*

R1: 3. As noted in the previous round of reviews, I recommend not using digital elevation model (DEM) when considering both elevation and depth like in this study. Since the data include cottages and vegetation, I believe that the most appropriate term would be Digital Surface Model (DSM), as opposed to Digital Terrain Model (DTM) that represents a "bare-earth" model.
> *We acknowledge the point that the elevation model includes cottages and vegetation, hence it could be called a DSM. However, after many considerations we have decided to maintain the term DEM, because this study generally focuses on the terrain/morphology, hence DSM will not be the appropriate term to use in this context. Only one figure visualizes the cottages and vegetation in the northern part on Fanø, and this part is not used in the geomorphometric and morphological analysis and classification. Obviously, DTM would be an incorrect term due to the cottages and vegetation, and therefore we have settled on the broader DEM term, which we find to encompass all the different environments in the elevation model.*
> *Regarding the use of "elevation" for topography and bathymetry, we have investigated the issue and we are confident that it can be used with positive values above sea level and negative values below, e.g. ESRI defines "elevation" as:*
> *"The vertical distance of a point or object above or below a reference surface or datum (generally mean sea level)"*
> *([http://support.esri.com/other-resources/gis-dictionary/search/elevation](http://support.esri.com/other-resources/gis-dictionary/search/elevation))*
> *In comparison they define "altitude" as:*
> *"The height or vertical elevation of a point above a reference surface. Altitude measurements are usually based on a given reference datum, such as mean sea level"*
> *([http://support.esri.com/other-resources/gis-dictionary/search/altitude](http://support.esri.com/other-resources/gis-dictionary/search/altitude))*
> *We also found that "DEM" is an often used term for bathymetric or topobathymetric elevation models in the published literature (Chust et al., 2010; Coleman et al., 2011; Fernandez-Diaz et al., 2014; Finkl et al., 2005; Galparsoro et al., 2013; Pe'eri and Long, 2011; Wedding et al., 2008).*
> *Based on the arguments above, we decided to keep the DEM-term, and to replace "altitude" with "elevation" throughout the text.*

R1: 4. Also noted before: "landscape" has different meanings depending on the field of study (e.g. in landscape ecology or remote sensing). I recommend removing the two instances on page 17 and replacing them by "terrain".

*Landscape has been changed to terrain as suggested.*

R1: 5. Another one noted before: change "small-scale" and "large-scale" for "fine-scale" and "broad-scale" throughout the text and in figures (e.g. Fig. 4). The formers have different meanings in cartography and other fields than in this study. The latters are less ambiguous.

*Small- and large-scale have been changed to fine- and broad-scale throughout the manuscript as suggested.*

R1: 6. On page 8 (lines 3 to 14), this should be moved to the methods section below, likely between lines 19 and 20 of that same page. At line 9, I would change the first "surface" for a word like "top" to make it clearer, e.g. "…located in the river with its top just below the water surface."

*Corrections have been done as suggested.*

R1: 7. I understand how the method that is described in this manuscript is more transparent, reproducible and user-friendly than current alternatives. However, I am unsure of the level of reproducibility for two reasons. First, while the authors use RiHYDRO, HydroFusion and LiDAR Survey Studio as examples for describing the lack of transparency in available software (p. 6), the proposed method still requires many software that may not be widely available for all users and are not necessarily more transparent (e.g. RiPROCESS, HydroVish, Fledermaus, MATLAB). Second, many steps involve manual processing (e.g. filtering, extracting the shallow surface) or subjective decisions (e.g. parameters for automatic filtering, classification trees, values of 4 for standard deviation to differentiate between features). I appreciate that the authors mention these limitations (e.g. p. 11 and 28), however I believe that care should be used when making claims like at page 22 ("it is open to the public"), especially since the software used may not be accessible to the public. I wouldn't go as far as suggesting the removal of mentions of "reproducible", but I would be curious to see if this issue will also be of concern to other reviewers or the editor.

*We acknowledge the point that we use software which is not accessible to the public; however, when using these software we describe how they work so there is no "black box". In this way the processing method is not software specific. In principle our method could be reproduced in other software packages, or one could develop own software using the proposed method. Therefore, we have changed the phrasing "open to the public", but we have not removed mentions of "reproducibility".*

R1: 8. On page 13, line 11, what do the authors mean by "only taking the top 95-100% of water points into account"? If 100% of the points are considered, then they are all accounted for and can't possibly increase reliability. Please clarify.

*We agree. It should be 1-5%. We have changed this accordingly.*

R1: 9. On page 13, lines 16 to 31 are confusing. I am unsure what is the relationship between the 2 x 2 m water surface and the 0.5 x 0.5 m surface. I understand that the 2 x 2 m was built to remove outliers, but what is it used for then? This section requires rewording or clarifications.

*We have rephrased this section in order to clarify the use of a 2 x 2 m and a 0.5 x 0.5 m grid, respectively.*

R1: 10. On pages 16 and 17, please specify what the standard deviation represents (for the BPI). Is it the standard deviation of depth/elevation values within the window of analysis?

*Yes, it is the STD of the altitude within the window of analysis. We have rephrased the paragraph to clarify this.*

R1: 11. On page 17 (lines 3-4), I do not understand what the authors mean by "the altitude was exaggerated 10 times before the classification, to enable the BTM to detect the shapes of the landscapes". A vertical exaggeration in the visual representation of data would not change the altitude (depth or elevation) values and thus would not impact the results. If this is the case, then this sentence is unnecessary and can be removed. However, if the altitude values were actually altered and multiplied by a factor of 10, I am not sure if the analysis is still valid, although it would likely not change the relative values of pixels and still identify peaks and pits (but maybe reduce the amount of flat areas?). This needs to be clarified.

*The actual values of the altitude were exaggerated by a factor of 10 because without exaggeration the BTM-tool did not show meaningful results with respect to the classification of crests and depressions. We have rephrased the paragraph to clarify this. We believe that it is still a valid analysis because the relative difference between cell values has not been changed. When it comes to slopes and flats, the input parameter was the slope of the actual altitudes (not the exaggerated), so the amount of flat areas is not affected by the exaggeration.*

R1: 12. Page 17, line 5: "the best results" based on what? Visual interpretation? Was it a subjective decision? Please specify.

*Yes, it was based on visual interpretation. We have clarified this.*

R1: 13. In Figure 5 and associated text, is everything >0.94 m really a beach dune? Weren't there cottages and other features? Would another term be more appropriate and all-encompassing of features that were above the water level?

*We are generally focusing on the natural environment, and therefore, we have excluded the northern part with cottages in the classification analyses. This is now clarified in the paper. Besides, Beach Dunes are not only > 0.94 m: The DEM also has to be 0.8 m above the "smooth DEM" as shown in Figure 5.*

R1: 14. On page 18, I believe that the window sizes are wrong. They would only make sense if a radius was used instead of a window of analysis. Window sizes need to be odd numbers and based on the pixel size (0.5 m), the window could not be of 100 m or 250 m wide. Please revise these measurements, indicate whether a window of analysis (square) was used or a radius (circle), and whether these numbers are the numbers of pixels of the window or the actual area covered by the window (in either cases it should be an odd number).

*You are absolutely right! We actually used 99.5 m and 249.5 m in square windows in the analysis, but we presented these as round numbers, i.e. 100 m and 250 m, to make it more comprehendible for the reader. We have now also added the exact numbers.*

R1: 15. For your information, standard deviation (cf. classification trees) is used as a measure of rugosity in geomorphometry, so when the authors used it to distinguish between bars and larger features, they used a measure of broad-scale rugosity.

*Yes, you are absolutely right, thanks. Rugosity is to some extent a "problematic" term, as it is defined and quantified differently in different disciplines, like the case of landscape.*

R1: 16. Page 18, lines 29-30: "4 were found to be a suitable ratio threshold". Based on what (e.g. visual interpretation, etc.)?

*Yes it was based on visual interpretation. We have clarified this.*

R1: 17. In Figure 10 (B-C-D), I would add a category in the legend to characterize the grey areas (that I assume are no data, i.e. the areas that did not correspond to any of the criteria in the decision/classification tree). This category could be named "Transition zones".

*We have termed this category "Unclassified".*

R1: 18. On page 25, lines 4 to 17 are repetitive to the methods section (p. 9). I would bring back the description of the environmental conditions at time of survey in the methods section, and keep the discussion on their implications at p. 25.

*Rearrangements have been made as suggested.*

R1: 19. The use of English language could be improved before publication, although it is not a big concern at the moment as the text remains clear. For instance, p. 10 line 21 should read "Steps 5-8 represent" instead or "Step 5-8 represents", p. 12 line 14 "currently" should be removed since it was at time of survey, p. 18 line 29 "was" instead of "were", p. 29 line 18 "are" instead of "is", etc.. These are only a few examples of where corrections should be made. Also, in some parts of the text that describe the methods and results, the past tense should be used.

*Thanks for your specific suggestions; additionally, we have improved the language, including the issues related to past and present tense.*

**Authors' response to Report #1 of Referee #2**

*Authors' response is italicized.*

R2: The authors had made adequate revisions and the quality has been improved largely. However, there are still some aspects to improve.

R2: 1. The research content is abundant, which can be divided into two papers: one for Processing and performance of topobathymetric LiDAR data, the other one for geomorphometric and morphological classification. Because of the abundant content, the aim of this research is difficult to focus.

> *Our original idea was, as you have suggested, dividing our study into two papers, which is why the paper at the first submission included only the Processing and performance of topobathymetric LiDAR. However; the first round of reviews, and complimented by the Editor, highlighted the necessity of including a morphological quantitative analysis in order to place the technical part of the study in context and to demonstrate the application of the topobathymetric LiDAR data. We acknowledged this; and in our response to the first round of reviews we stated how we would approach this, which was then realised by adding the geomorphometric and morphological analysis in the revised manuscript. In retrospect we truly agree with the referee in review report 2 of the second round of reviews that the application (i.e. the morphological characterization) makes it a more complete paper. Moreover, we believe that it is very valuable to present the complete processing line from raw topobathymetric LiDAR data to automatic landform classification based on geomorphometric analyses. Finally, we sincerely believe that we have developed novel methods for processing green LiDAR data and for classifying morphological units in coastal tidal environments using geomorphometry.*

R2: 2. The presentation can be improved in some aspects. For example, the "introduction" section is too long, and Page 4 can be placed into "methodology" section. In addition, Section 3.2 can be made into a diagram.

> *We have aimed at improving the presentation of the study and its findings: We have shortened the introduction section; this includes moving the description of topobathymetric LiDAR to the methods section as suggested. Section 3.2 is already visualized in the quite comprehensive Figure 3, which functions both as a flow diagram of the processing steps as well as a visualization of the individual processing steps. So we believe the referee suggestion has been acknowledged in the paper.*

R2: 3. Because of abundant content, the figures is too much.

> *We acknowledge that the number of figures may seem relatively high but we find the figures relevant to aid the reader and to convey our findings. However, we suggest excluding Fig. 12, which shows the spatial coverage of the dead-zone at different water levels. We have removed this figure from the paper and made the required changes in the text.*

R2: 4. The geomorphometric classification is not mentioned in the "conclusion" section.

> *The results of the geomorphometric classification were not included in the conclusions, as this was considered a step on the way towards the final morphological classification. However, we truly acknowledge that the geomorphometric classification constitutes one of the central parts of the study; hence we have included it in the conclusions as suggested.*

R2: Overall, I do not think one paper should put so many issues. I suggest the authors can focus on one or two issues and illustrate deeply and concisely.

> *Please refer to the history of the paper outlined under point 1.*

**1** **Processing and performance of topobathymetric LiDAR**

**2** **data for geomorphometric and morphological**

**3** **classification in a high-energy tidal environment**

**4**

[revised manuscript text omitted]